

PeerJ Hubs
Published on behalf of

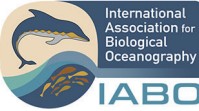
International Association for Biological Oceanography
IABO

# Stirring up the muck: the systematics of soft-sediment Fionidae (Nudibranchia: Aeolidina) from the tropical Indo-Pacific

Ashley Y. Kim[1,2], Samantha A. Donohoo[1,3] and Terrence M. Gosliner[1]

[1] Department of Invertebrate Zoology and Geology, California Academy of Sciences, San Francisco, California, United States
[2] School of Biological Sciences, Section of Ecology, Behavior, and Evolution, University of California, San Diego, La Jolla, California, United States
[3] School of Fisheries, Aquaculture, and Aquatic Sciences, Auburn University, Auburn, Alabama, United States

Corresponding author
Terrence M. Gosliner,
tgosliner@calacademy.org

## ABSTRACT

The tropical Indo-Pacific aeolid nudibranchs of the Fionidae are poorly known and have not been studied in a concerted manner. Many undescribed species are found throughout the Indian and Pacific Oceans and are concentrated in the Coral Triangle. With the recent publication of a revised systematic arrangement of the Fionidae, documentation and description of new taxa of Fionidae is especially warranted. Here we describe *Tenellia bughaw* Kim & Gosliner sp. nov., *Tenellia puti* Kim & Gosliner sp. nov., *Tenellia nakapila* Kim & Gosliner sp. nov., *Abronica payaso* Kim & Gosliner, sp. nov. and *Abronica turon* Gosliner & Kim sp. nov. from the waters of the Coral Triangle. Their phylogenetic placement in Fionidae is reviewed using three genes: cytochrome oxidase I (COI), 16s rRNA, and histone 3 (H3) in a Bayesian and maximum likelihood framework. A redescription of *Tenellia yamasui* (Hamatani, 1993) is also provided to clarify its distinctiveness from *T. bughaw* and *T. puti*. This study confirms that all four known species of *Abronica* are characterized by having an acutely pointed curved penial stylet, thus confirming a unique morphological synapomorphy for members of this genus. A discussion on conflicts in the classification of fionid aeolid nudibranchs addresses concerns with extreme splitting based on novel innovations that emphasize evolutionary novelty over phyletic kinship. Instead, a more conservative approach is suggested, especially within the context of taxa that still have much undocumented primary diversity.

# INTRODUCTION

The Indo-Pacific tropics harbor the richest marine biota within the largest part of the world's oceans, while the Coral Triangle supports the greatest diversity of species (*Roberts et al., 2002*). The documented diversity of nudibranchs and other heterobranch gastropods exceeds 1,400 species from the Indo-Pacific including approximately 120 species of Fionidae (previously identified as Eubranchidae, Calmidae, Tergipedidae, and Fionidae). Approximately 105 of the 120 fionid species likely represent undescribed taxa. Previously, *Cella et al. (2016)*, based on molecular phylogenetic and morphological data, revised the

systematics of the Tergipedidae to include Tergipedidae, Eubranchidae, Calmidae, and Fionidae all within the Fionidae, the oldest name for this monophyletic group. This molecular phylogeny includes several undescribed species within the genera *Tenellia* A. Costa, 1866, *Cuthonella* Bergh, 1884, *Abronica Cella et al., 2016*, and *Eubranchus* Forbes, 1838. Since then, additional species have been described and revisions to the taxonomy of this clade have been proposed (*Korshunova, Martynov & Picton, 2017*; *Korshunova et al., 2018a*, *2018b*; *Ekimova, Deart & Schepetov, 2019*; *Fritts-Penniman et al., 2020*; *Martynov et al., 2020*; *Korshunova et al., 2023b*). Most notably, *Korshunova, Martynov & Picton (2017)* divided the single family Fionidae (of *Cella et al. (2016)*) into seven families (Eubranchidae, Calmidae, Cuthonellidae, Cuthonidae, Tergipedidae, Fionidae, and Trinchesiidae) with five of these families including only a single genus and a handful of species. This was based largely on the phylogeny of *Cella et al. (2016)* and the addition of one additional species. Within their Fionoidea, *Korshunova, Martynov & Picton (2017)* included the following genera within Trinchesiidae: *Catriona* Winckworth, 1941, *Diaphoreolis* Iredale & O'Donoghue, 1923, *Phestilla* Bergh, 1874, *Tenellia*, *Trinchesia* Ihering, 1879, and *Zelentia Korshunova, Martynov & Picton, 2017*. In this article, we describe three new species of *Tenellia* based on a larger generic concept we follow and explain. Additionally, *Korshunova, Martynov & Picton (2017)*, erected two additional families with Fionoidea: Abronicadae and Muraniidae, both of which contain a single genus (*Abronica* and *Murmania*, respectively) and a small number of species in each genus (1 to 3). In the case of *Abronica*, we describe two additional species in this article. *Korshunova, Martynov & Picton (2017)*, contended that there are clear morphological and molecular differences between the genera they recognized; however, we provide an alternative view in this article of why we have chosen a different approach. As a result, there is little consensus regarding the taxonomy of Fionidae (*sensu lato*) and this article discusses the merits of the two primary conflicting perspectives. Most taxonomic work has focused on cold-temperate and boreal taxa from the northern hemisphere with the exceptions of *Cella et al. (2016)*, *Fritts-Penniman et al. (2020)*, *Mehrotra et al. (2020)*, *Hu et al. (2020)*, *Wang et al. (2020)*, and *Mehrotra et al. (2024)*, with little attention directed at the undocumented tropical diversity of the Indian and Pacific Oceans. Most of these articles dealt exclusively with coral-eating species rather than a broad spectrum of taxa.

In *Rudman (2002)*, Rudman identified what he considered to be a highly variable species, *Cuthona yamasui Hamatani, 1993* (originally described from Japan), based on specimens from Australia and noted that specimens depicted by *Koehler (1998)* and *Picton (2002)* also likely represented this species. Based on this identification, *Gosliner, Behrens & Valdés (2008*: 360, top four photos) illustrated four individuals of this species. More recently, *Gosliner, Valdés & Behrens (2015*: 347, top two photos) indicated that a different species actually represented the true *C. yamasui* and that the specimens they illustrated in 2008 actually represented two distinct species, *Cuthona* sp. 13 and *Cuthona* sp. 14, respectively. The placement of this species in *Cuthona* was based on the fact that the vast majority of members in this family were lumped in the genus *Cuthona*. With the publication of the phylogeny of *Cella et al. (2016)* the first molecular phylogeny of the group began producing a better understanding of the systematics of the larger clade. These

two species were nested in a large clade of *Tenellia* species in *Cella et al. (2016)* and were listed as *Tenellia* sp. E and *Tenellia* sp. F, based on molecular data. This article describes these two species and clearly differentiates them from *T. yamasui*. Two new species of *Abronica*, formerly *Abronica* sp. 6 and *Abronica* sp. 7, and another undescribed species of *Tenellia* recently collected from the Philippines are also described.

## MATERIALS AND METHODS

### Molecular study

All specimens processed for molecular work were preserved in 95% EtOH. A total of 150 specimens, six newly sequenced and 144 with two or more genes previously published and available on GenBank from *Faucci, Toonen & Hadfield (2007)*, *Pola & Gosliner (2010)*, *Moore & Gosliner (2011)*, *Carmona et al. (2013)*, *Cella et al. (2016)*, *Korshunova et al. (2017, 2018b, 2019, 2020, 2023a)*, *Ekimova, Deart & Schepetov (2019)*, *Ekimova, Grishina & Nikitenko (2024)*, and *Mehrotra et al. (2020, 2024)* were used in the phylogenetic analyses. Seventeen new sequences were deposited on GenBank with the following accession numbers: partial fragments of the 16S ribosomal RNA gene (16S rRNA; PP759731–PP759736, PP751617), cytochrome C oxidase subunit I gene (COI; PP747810–PP747814, PP751619), histone H3 gene (H3; PP750950–PP750955, PP768732). Sampled specimens with previous and present species identifications, voucher numbers, locality information, GenBank accession numbers, and references are listed in Table 1. Members of Tritoniidae, Aeolidiidae, Babakinidae, Facelinidae, and Flabellinidae were used for outgroup comparisons based on molecular phylogenetic analysis by *Pola & Gosliner (2010)* and *Cella et al. (2016)*. Vouchers of newly sequenced specimens and holotypes are deposited at the California Academy of Sciences (CASIZ) and the National Museum of Philippines (NMP).

### DNA extraction, amplification, and Sanger sequencing

Genomic DNA from two mitochondrial genes, cytochrome oxidase I (COI) and 16S rRNA, and one nuclear gene, histone 3 (H3), were extracted, amplified using polymerase chain reaction (PCR) and universal primers, and sequenced on an ABI3130 Genetic Analyzer following protocols by Kristen Cella and Terrence M. Gosliner outlined in *Cella et al. (2016)* (Supplemental Material). Briefly, each PCR reaction used universal gene-specific primers: 16S rRNA (16S arL, 16S R; *Palumbi et al., 1991*); COI (HCO 2198, LCO 1490; *Folmer et al., 1994*); and H3 (H3 AF, H3 AR; *Colgan et al., 1998*) and gene-specific protocols which were run on a BioRad MyCycler ThermocyclerB (Bio-Rad Laboratories, Hercules, CA, USA). PCR amplification occurred in 25 μl reactions containing: 1 μl of genomic DNA template, 2.5 μl of 10× PCR buffer, 0.5 μl dNTPs (10 μM stock), 1.5 μl MgCl2 (25 μM stock), 1 μl of bovine serum albumin, 0.2 μl each primer (25 μM stock), 0.5 μl Taq-Apex (1.25 units/mL) and 17.6 μl of de-ionized water (ddH20). PCR protocols are as follows: for COI and 16S rRNA, an initial denaturing for 3 min at 94 °C, followed by 40 cycles of denaturing for 30 s at 94 °C, annealing for 30 s at 50–56 °C, and extension for 1 min at 72 °C with a final extension period of 5 min at 72 °C and for H3, an initial denaturing for 3 min at 94 °C, followed by 35 cycles of denaturing for 35 s at

**Table 1 Specimens sequenced with present study identifications, previous species identifications, voucher numbers, GenBank accession numbers, locality information, and sequencing references.** Dashes indicate missing sequences.

| Present study species identifications | Previous species identifications | Voucher | Locality | Accession numbers 16S | Reference COI | H3 | |
|---|---|---|---|---|---|---|---|
| Tritoniidae Lamarck, 1809 | | | | | | | |
| *Tritonicula pickensi* (Marcus and Marcus, 1967) | | CASIZ 175718 | Costa Rica | HM162642 | HM162717 | HM162549 | *Pola & Gosliner (2010)* |
| *Marionia distincta* Bergh, 1905 | | CASIZ 173317 | Philippines | HM162648 | HM162725 | HM162557 | *Pola & Gosliner (2010)* |
| Aeolididae Gray, 1827 | | | | | | | |
| *Aeolidia loui* Kienberger, Carmona, Pola, Padula, Gosliner, and Cervera, 2016 | | CASIZ 173369 | California | KY128766 | KY128974 | KY128561 | *Cella et al. (2016)* |
| Babakinidae Roller, 1973 | | | | | | | |
| *Babakina indopacifica* Gosliner, González-Duarte and Cervera, 2007 | | CASIZ 177458 | Philippines | HM162678 | HM162754 | HM162587 | *Pola & Gosliner (2010)* |
| Facelinidae Bergh, 1889 | | | | | | | |
| *Cratena pilata* (Gould, 1870) | | CASIZ 184187 | Massachusetts | KY128709 | – | KY128502 | *Cella et al. (2016)* |
| *Facelina bostoniensis* (Couthouy, 1838) | | CASIZ 184184 | New Hampshire | KY128837 | KY129046 | KY128632 | *Cella et al. (2016)* |
| *Godiva quadricolor* (Barnard, 1927) | | CASIZ 176385 | South Africa | HM162680 | HM162756 | HM162589 | *Pola & Gosliner (2010)* |
| *Phyllodesmium opalescens* Rudman, 1991 | | CASIZ 177311 | Philippines | HQ010518 | HQ010484 | HQ010449 | *Moore & Gosliner (2011)* |
| *Phyllodesmium parangatum* Ortiz & Gosliner, 2003 | | CASIZ 174440 | Philippines | KY128872 | KY129081 | – | *Cella et al. (2016)* |
| *Phyllodesmium* sp. | *Phyllodesmium* sp. A | CASIZ 177476 | Philippines | KY128873 | KY129082 | KY128666 | *Cella et al. (2016)* |
| *Phyllodesmium* sp. | *Phyllodesmium* sp. A | CASIZ 181302 | Philippines | KY128874 | KY129083 | KY128667 | *Cella et al. (2016)* |
| Flabellinidae Bergh, 1889 | | | | | | | |
| *Apata pricei* (*MacFarland, 1966*) | *Flabellina pricei* (*MacFarland, 1966*) | CASIZ 114776 | California | KY128851 | KY129060 | KY128645 | *Cella et al. (2016)* |
| *Apata* cf. *pricei* (*MacFarland, 1966*) | *Flabellina* sp. A | CASIZ 181322 | California | KY128843 | KY129052 | KY128637 | *Cella et al. (2016)* |
| *Coryphella gracilis* (Alder and Hancock, 1844) | *Flabellina gracilis* (Alder and Hancock, 1844) | CASIZ 183938 | Maine | KY128846 | KY129055 | KY128640 | *Cella et al. (2016)* |
| *Coryphella trilineata* O'Donoghue, 1921 | *Flabellina trilineata* (O'Donoghue, 1921) | CASIZ 179466 | California | KY128855 | KY129064 | KY128649 | *Cella et al. (2016)* |
| *Coryphella verrucosa* (M. Sars, 1829) | *Flabellina verrucosa* (M. Sars, 1829) | CASIZ 183939 | Maine | KY128856 | KY129065 | KY128650 | *Cella et al. (2016)* |

| Present study species identifications | Previous species identifications | Voucher | Locality | Accession numbers 16S | Reference COI | H3 | |
|---|---|---|---|---|---|---|---|
| *Coryphellina arveloi* (Ortea and Espinosa, 1998) | | CASIZ 179419 | Sao Tome and Principe | KY128840 | KY129049 | KY128634 | *Cella et al. (2016)* |
| *Coryphellina exoptata* (Gosliner and Willan, 1991) | *Flabellina exoptata* Gosliner and Willan, 1991 | CASIZ 178322 | Malaysia | KY128844 | KY129053 | KY128638 | *Cella et al. (2016)* |
| *Coryphellina lotos Korshunova et al., 2017* | *Flabellina rubrolineata* (O'Donoghue, 1929) | CASIZ 177287 | Philippines | KY128852 | KY129061 | KY128646 | *Cella et al. (2016)* |
| *Flabellinopsis iodinea* (J. G. Cooper, 1863) | *Flabellina iodinea* (J. G. Cooper, 1863) | CASIZ 181313a | California | KY128847 | KY129056 | KY128641 | *Cella et al. (2016)* |
| *Pacifia goddardi* (Gosliner, 2010) | *Flabellina goddardi* Gosliner, 2010 | CASIZ 182590 | California | KY128854 | KY129063 | KY128648 | *Cella et al. (2016)* |
| *Paraflabellina funeka* (Gosliner and Griffiths, 1981) | *Flabellina funeka* Gosliner and Griffiths, 1981 | CASIZ 176374 | South Africa | KY128845 | KY129054 | KY128639 | *Cella et al. (2016)* |
| *Samla bicolor* (Kelaart, 1858) | *Flabellina bicolor* (Kelaart, 1858) | CASIZ 177345 | Philippines | KY128841 | KY129050 | KY128635 | *Cella et al. (2016)* |
| *Samla bilas* (Gosliner and Willan, 1991) | *Flabellina bilas* Gosliner and Willan, 1991 | CASIZ 177355 | Philippines | KY128842 | KY129051 | KY128636 | *Cella et al. (2016)* |
| *Samla macassarana* (Bergh, 1905) | *Flabellina macassarana* Bergh, 1905 | CASIZ 181283 | Philippines | KY128850 | KY129059 | KY128644 | *Cella et al. (2016)* |
| *Ziminella salmonacea* (Couthouy, 1838) | *Flabellina salmonacea* (Couthouy, 1838) | CASIZ 183927 | Maine | KY128853 | KY129062 | KY128647 | *Cella et al. (2016)* |
| Fionidae Gray, 1857 | | | | | | | |
| *Abronica abronia* (*MacFarland, 1966*) | *Cuthona abronia* (*MacFarland, 1966*) | CASIZ 174485 | California | KY128712 | KY128917 | KY128504 | *Cella et al. (2016)* |
| *Abronica abronia* (*MacFarland, 1966*) | *Cuthona abronia* (*MacFarland, 1966*) | CASIZ 179463a | California | KY128713 | KY128918 | KY128505 | *Cella et al. (2016)* |
| *Abronica abronia* (*MacFarland, 1966*) | *Cuthona abronia* (*MacFarland, 1966*) | CASIZ 179463b | California | KY128714 | KY128919 | KY128506 | *Cella et al. (2016)* |
| *Abronica abronia* (*MacFarland, 1966*) | *Cuthona abronia* (*MacFarland, 1966*) | CASIZ 179463c | California | KY128715 | KY128920 | KY128507 | *Cella et al. (2016)* |
| *Abronica abronia* (*MacFarland, 1966*) | *Cuthona abronia* (*MacFarland, 1966*) | CASIZ 181319 | California | KY128716 | KY128919 | KY128508 | *Cella et al. (2016)* |
| *Abronica payaso* Kim & Gosliner sp. nov | *Cuthona* sp. 6 | CASIZ 177350 | Philippines | KY128780 | KY128988 | KY128575 | *Cella et al. (2016)* |
| *Abronica payaso* Kim & Gosliner sp. nov | *Cuthona* sp. 6 | CASIZ 177353 | Philippines | KY128781 | KY128989 | KY128576 | *Cella et al. (2016)* |
| *Abronica payaso* Kim & Gosliner sp. nov | *Cuthona* sp. 6 | NMP 041348 /CASIZ 177417 | Philippines | KY128782 | KY128990 | KY128577 | *Cella et al. (2016)* |
| *Abronica purpureoanulata* (*Baba, 1961*) | *Cuthona purpureoanulata* (*Baba, 1961*) | CASIZ 177607 | Philippines | KY128762 | KY128970 | KY128557 | *Cella et al. (2016)* |
| *Abronica turon* Kim & Gosliner sp. nov | *Cuthona* sp. 7 | CASIZ 179946 | Hawaii | KY128783 | KY128991 | KY128578 | *Cella et al. (2016)* |

(Continued)

| Present study species identifications | Previous species identifications | Voucher | Locality | Accession numbers 16S | Reference | | |
|---|---|---|---|---|---|---|---|
| | | | | | COI | H3 | |
| *Calma glaucoides* (Alder and Hancock, 1854) | | GnM9030 | Sweden | KY128705 | KY128913 | – | *Cella et al. (2016)* |
| *Calma gobioophaga* Calado and Urgorri, 2002 | | MNCN 408 | | HG810890 | HG810896 | – | *Cella et al. (2016)* |
| *Cuthona divae* (Er. Marcus, 1961) | | CASIZ 179470b | California | KY128738 | KY128943 | KY128531 | *Cella et al. (2016)* |
| *Cuthona divae* (Er. Marcus, 1961) | | CASIZ 179477 | California | KY128739 | KY128944 | KY128532 | *Cella et al. (2016)* |
| *Cuthona divae* (Er. Marcus, 1961) | | CASIZ 181316 | California | KY128741 | KY128946 | KY128534 | *Cella et al. (2016)* |
| *Cuthona nana* (Alder and Hancock, 1842) | | CASIZ 182700 | New Hampshire | KY128754 | KY128961 | KY128548 | *Cella et al. (2016)* |
| *Cuthona nana* (Alder and Hancock, 1842) | | AC14-10 | Sea of Japan | KY128755 | KY128962 | KY128549 | *Cella et al. (2016)* |
| *Cuthona nana* (Alder and Hancock, 1842) | | AC22-14 | Netherlands | KY128756 | KY128964 | KY128551 | *Cella et al. (2016)* |
| *Cuthonella ainu* *Korshunova et al., 2020* | | ZMMU:Op-618 | Russia: Matua | MW158746 | MW150866 | MW158334 | *Korshunova et al. (2020)* |
| *Cuthonella anastasia* *Ekimova, Grishina & Nikitenko, 2024* | | MIMB48075 | Russia: Sea of Okhotsk | – | PP400675 | PP412173 | *Ekimova, Grishina & Nikitenko, 2024* |
| *Cuthonella benedykti* *Korshunova et al., 2020* | | ZMMU:Op-194 | Russia: Kamachatka | MW158747 | MW150867 | MW158322 | *Korshunova et al. (2020)* |
| *Cuthonella cocoachroma* (Williams and Gosliner, 1979) | *Cuthona cocoachroma* *Williams & Gosliner, 1979* | CASIZ 179471 | California | KY128720 | KY128925 | KY128513 | *Cella et al. (2016)* |
| *Cuthonella concinna* (Alder and Hancock, 1843) | *Cuthona concinna* (Alder and Hancock, 1843) | CASIZ 182702 | Maine | KY128729 | KY128934 | KY128522 | *Cella et al. (2016)* |
| *Cuthonella concinna* (Alder and Hancock, 1843) | *Cuthona concinna* (Alder and Hancock, 1843) | CASIZ 181522b | Alaska | KY128727 | KY128932 | KY128520 | *Cella et al. (2016)* |
| *Cuthonella concinna* (Alder and Hancock, 1843) | *Cuthona concinna* (Alder and Hancock, 1843) | CASIZ 179469 | California | KY128719 | KY128924 | KY128512 | *Cella et al. (2016)* |
| *Cuthonella concinna* (Alder and Hancock, 1843) | *Cuthonella marisalbi* (Roginskaya, 1963) | WS3446 | Russia: White Sea | KY128806 | KY129015 | KY128601 | *Cella et al. (2016)* |
| *Cuthonella denbei* *Korshunova et al., 2020* | | ZMMU:Op-673 | Russia: Matua | MW158748 | MW150868 | MW158331 | *Korshunova et al. (2020)* |
| *Cuthonella georgstelleri* *Korshunova et al., 2020* | | ZMMU:Op-670 | Russia: Matua | MW158741 | MW150861 | MW158330 | *Korshunova et al. (2020)* |
| *Cuthonella hiemalis* (Roginskaya, 1987) | *Cuthona hiemalis* Roginskaya, 1987 | WS3440 | Russia: White Sea | KY128801 | KY129009 | KY128597 | *Cella et al. (2016)* |

| Present study species identifications | Previous species identifications | Voucher | Locality | Accession numbers 16S | Reference | | |
|---|---|---|---|---|---|---|---|
| | | | | | COI | H3 | |
| *Cuthonella osyoro* (Baba, 1940) | *Cuthona osyoro* Baba, 1940 | ZMMU:Op-606 | Russia: Sea of Japan | MW158735 | MW150855 | MW158325 | *Korshunova et al. (2020)* |
| *Cuthonella punicea* (Millen, 1986) | *Cuthona punicea* Millen, 1986 | ZMMU:Km-766 | Canada | MW158738 | MW150858 | MW158337 | *Korshunova et al. (2020)* |
| *Cuthonella sandrae* *Korshunova et al., 2020* | | ZMMU:Op-671 | Russia: Matua | MW158743 | MW150863 | MW158333 | *Korshunova et al. (2020)* |
| *Cuthonella vasentsovichi* *Korshunova et al., 2020* | | ZMMU:OP-738 | Russia: Matua | MW158744 | MW150864 | MW158335 | *Korshunova et al. (2020)* |
| *Eubranchus alexeii* (Martynov, 1998) | *Aenigmastyletus alexeii* Martynov, 1998 | WS3432 | Sea of Japan | KY128692 | KY128900 | KY128487 | *Cella et al. (2016)* |
| *Eubranchus farrani* (Alder and Hancock, 1844) | *Amphorina farrani* (Alder & Hancock, 1844) | GnM9093 | Sweden | KY128819 | KY129028 | KY128614 | *Cella et al. (2016)* |
| *Eubranchus mandapamensis* (Rao, 1968) | | CASIZ 177750a | Philippines | KY128826 | KY129035 | KY128621 | *Cella et al. (2016)* |
| *Eubranchus odhneri* (Derjugin & Gurjanova, 1926) | | WS3435 | White Sea | KY128695 | KY128903 | KY128490 | *Cella et al. (2016)* |
| *Eubranchus olivaceus* (O'Donoghue, 1922) | *Eubranchus rupium* (Møller, 1842) | CASIZ 181133 | California | KY128828 | KY129037 | KY128623 | *Cella et al. (2016)* |
| *Eubranchus pallidus* (Alder & Hancock, 1842) | *Amphorina pallida* (Alder & Hancock, 1842) | WS3454 | Barents Sea | KY128824 | KY129033 | KY128619 | *Cella et al. (2016)* |
| *Eubranchus* sp. 23 | *Eubranchus* sp. 3 | CASIZ 181292a | Philippines | KY128831 | KY129040 | KY128626 | *Cella et al. (2016)* |
| *Eubranchus rupium* (Møller, 1842) | | CASIZ 183925 | Maine | KY128825 | KY129034 | KY128620 | *Cella et al. (2016)* |
| *Eubranchus rupium* (Møller, 1842) | | WS3459 | White Sea | KY128862 | KY129070 | KY128655 | *Cella et al. (2016)* |
| *Eubranchus scintillans* Grishina, Schepetov & Ekimova, 2022 | *Eubranchus exiguus* (Alder and Hancock, 1848) | GnM9092 | Scotland | KY128820 | KY129029 | KY128615 | *Cella et al. (2016)* |
| *Eubranchus tricolor* Forbes, 1838 | | GnM9096 | Sweden | KY128823 | KY129032 | KY128618 | *Cella et al. (2016)* |
| *Fiona pinnata* (Eschscholtz, 1831) | | CASIZ 179238 | Vanuatu | KY128838 | KY129047 | KY128486 | *Cella et al. (2016)* |
| *Fiona pinnata* (Eschscholtz, 1831) | | MNCN/ADN 51997 | Morocco | JX087492 | JX087558 | JX087628 | *Carmona et al. (2013)* |
| *Murmania antiqua* Martynov, 2006 | | WS3455 | Kara Sea | KY128857 | KY129066 | KY128651 | *Cella et al. (2016)* |
| *Rubramoena amoena* (Alder & Hancock, 1845) | *Cuthona amoena* (Alder & Hancock, 1845) | GnM9098 | Great Britain | KY128696 | KY128904 | KY128491 | *Cella et al. (2016)* |

(Continued)

| Present study species identifications | Previous species identifications | Voucher | Locality | Accession numbers 16S | Reference | |
|---|---|---|---|---|---|---|
| | | | | | COI | H3 |
| *Rubramoena rubescens* (Picton & Brown, 1978) | *Cuthona rubescens* Picton & Brown, 1978 | GnM9102 | Great Britain | KY128710 | KY128916 | KY128503 | *Cella et al. (2016)* |
| *Tenellia adspersa* (Nordmann, 1845) | | CASIZ 184191 | New Hampshire | KY128876 | KY129085 | KY128668 | *Cella et al. (2016)* |
| *Tenellia arnoldi* (Mehrotra & Caballer, 2024) | *Phestilla arnoldi* Mehrotra & Caballer, 2024 | RM-2023 | Thailand | OQ772262 | OQ745796 | OQ789632 | *Mehrotra et al. (2024)* |
| *Tenellia aurantia* (Alder & Hancock, 1842) | *Catriona aurantia* (Alder & Hancock, 1842) | ZMMU:Op-545 | Norway | MF523458 | KY985467 | MG386404 | *Korshunova et al. (2017)* |
| *Tenellia bughaw* Kim & Gosliner sp. nov. | *Trinchesia yamasui* (*Hamatani, 1993*) | CASIZ 176737 | Malaysia | KY128791 | KY128999 | KY128586 | *Cella et al. (2016)* |
| *Tenellia bughaw* Kim & Gosliner sp. nov. | *Trinchesia yamasui* (*Hamatani, 1993*) | CASIZ 176739a | Malaysia | KY128792 | KY129000 | KY128587 | *Cella et al. (2016)* |
| *Tenellia bughaw* Kim & Gosliner sp. nov. | *Trinchesia yamasui* (*Hamatani, 1993*) | CASIZ 176739b | Malaysia | KY128793 | KY129001 | KY128588 | *Cella et al. (2016)* |
| *Tenellia bughaw* Kim & Gosliner sp. nov. | *Trinchesia yamasui* (*Hamatani, 1993*) | CASIZ 181298 | Philippines | KY128798 | KY129006 | KY128593 | *Cella et al. (2016)* |
| *Tenellia bughaw* Kim & Gosliner sp. nov. | *Trinchesia yamasui* (*Hamatani, 1993*) | CASIZ 177552 | Philippines | KY128795 | KY129003 | KY128590 | *Cella et al. (2016)* |
| *Tenellia caerulea* (Montagu, 1804) | *Trinchesia caerulea* (Montagu, 1804) | CASIZ 185199 | Spain | KY128717 | KY128922 | KY128510 | *Cella et al. (2016)* |
| *Tenellia chaetopterana* (Ekimova, Deart & Schepetov, 2017) | *Phestilla chaetopterana* (Ekimova, Deart & Schepetov, 2017) | WS8071 | Vietnam | MF458306 | MF458312 | MF458309 | *Ekimova, Deart & Schepetov (2019)* |
| *Tenellia columbiana* (O'Donoghue, 1922) | *Catriona columbiana* (O'Donoghue, 1922) | ZMMU:Op-486 | Sea of Japan | OP070020 | OP062245 | OP185385 | *Korshunova et al. (2023a)* |
| *Tenellia cuanensis* (*Korshunova et al., 2019*) | *Trinchesia cuanensis* *Korshunova et al., 2019* | GnM9054 | United Kingdom | MK587935 | MK587920 | MK587905 | *Korshunova et al. (2019)* |
| *Tenellia diljuvia* (*Korshunova et al., 2019*) | *Trinchesia diljuvia* *Korshunova et al., 2019* | ZMMU:Op-642 | Russia: Black Sea | MK587933 | MK587917 | MK587903 | *Korshunova et al. (2019)* |
| *Tenellia flavovulta* (*MacFarland, 1966*) | *Diaphoreolis flavovulta* (*MacFarland, 1966*) | CASIZ 181132 | California | KY128745 | KY128950 | KY128538 | *Cella et al. (2016)* |
| *Tenellia foliata* (Forbes & Goodsir, 1839) | *Trinchesia foliata* (Forbes & Goodsir, 1839) | GnM9100 | Ireland | KY128704 | KY128912 | KY128499 | *Cella et al. (2016)* |
| *Tenellia fulgens* (*MacFarland, 1966*) | *Zelentia fulgens* (*MacFarland, 1966*) | CASIZ 185194 | California | KY128747 | KY128952 | KY128540 | *Cella et al. (2016)* |
| *Tenellia fuscostriata* (*Hu et al., 2020*) | *Phestilla fuscostriata* *Hu et al., 2020* | JH-2020-IsoA | China | MN065807 | MN065805 | MN065809 | GenBank |
| *Tenellia gotlandica* Lundin, Malmberg, Martynov & Korshunova, 2022 | | GnM9960 | Sweden | OP070013 | OP062247 | OP185376 | *Korshunova et al. (2023a)* |

| Present study species identifications | Previous species identifications | Voucher | Locality | Accession numbers 16S | Reference COI | H3 | |
|---|---|---|---|---|---|---|---|
| *Tenellia gymnota* (Couthouy, 1838) | *Catriona gymnota* (Couthouy, 1838) | CASIZ 184188 | New Hampshire | KY128700 | KY128908 | KY128495 | *Cella et al. (2016)* |
| *Tenellia kishiwadensis* (Martynov, Korshunova, Lundin & Malmberg, 2022) | *Catriona kishiwadensis* Martynov, Korshunova, Lundin & Malmberg, 2022 | KSNHM-M10590.3 | Japan | OP070008 | OP062238 | OP185381 | *Korshunova et al. (2023a)* |
| *Tenellia lagunae* (O'Donoghue, 1926) | *Diaphoreolis lagunae* (O'Donoghue, 1926) | CASIZ 175583 | California | KY128748 | KY128955 | KY128542 | *Cella et al. (2016)* |
| *Tenellia lenkae* (Martynov, 2002) | *Trinchesia lenkae* Martynov, 2002 | AC17-19 | Sea of Japan | KY128884 | KY129093 | KY128676 | *Cella et al. (2016)* |
| *Tenellia lucerna* (Korshunova, Martynov, Lundin & Malmberg, 2022) | *Catriona lucerna* Korshunova, Martynov, Lundin & Malmberg, 2022 | ZMMU:Op-789 | Vietnam | OP070012 | OP062243 | OP185383 | *Korshunova et al. (2023a)* |
| *Tenellia lugubris* (Bergh, 1870) | *Phestilla lugubris* (Bergh, 1870) | CASIZ 177437 | Philippines | KY128866 | KY129075 | KY128660 | *Cella et al. (2016)* |
| *Tenellia* cf. *maua* (Ev. Marcus and Er. Marcus, 1960) | *Catriona* cf. *maua* Ev. Marcus and Er. Marcus, 1960 | CASIZ 179403 | Sao Tome and Principe | KY128697 | KY128905 | – | *Cella et al. (2016)* |
| *Tenellia melanobrachia* (Bergh, 1874) | *Phestilla melanobrachia* Bergh, 1874 | CASIZ 167974a | Papua New Guinea | KY128867 | KY129076 | KY128661 | *Cella et al. (2016)* |
| *Tenellia midori* (Martynov, Sanamyan & Korshunova, 2015) | *Trinchesia midori* Martynov, Sanamyan & Korshunova, 2015 | ZMMU:Op-830 | Japan | OQ779044 | OQ779512 | OQ787050 | *Korshunova et al. (2023a)* |
| *Tenellia morrowae* (Korshunova, Picton, Furfaro, Mariottini, Pontes, Prkić, Fletcher, Malmberg, Lundin & Martynov, 2019) | *Trinchesia morrowae* Korshunova, Picton, Furfaro, Mariottini, Pontes, Prkić, Fletcher, Malmberg, Lundin & Martynov, 2019 | ZMMU:Op-651 | Spain | MK587938 | MK587924 | MK587908 | *Korshunova et al. (2019)* |
| *Tenellia nakapila* Kim & Gosliner sp. nov. | *Tenellia* sp. 20 | CASIZ 208579 | Philippines | PP759736 | PP747814 | PP750955 | Present study |
| *Tenellia nakapila* Kim & Gosliner sp. nov. | *Tenellia* sp. 20 | NMP 041347 /CASIZ 202110 | Philippines | PP759734 | - | PP750953 | Present study |
| *Tenellia nakapila* Kim & Gosliner sp. nov. | *Tenellia* sp. 20 | CASIZ 217303 | Philippines | PP759735 | PP747813 | PP750954 | Present study |
| *Tenellia nepunicea* (*Korshunova et al., 2018b*) | *Zelentia nepunicea* *Korshunova et al., 2018b* | ZMMU:Op-627 | USA: Washington | MH614976 | MH614985 | MH614996 | *Korshunova et al. (2018b),* |
| *Tenellia ninel* (*Korshunova, Martynov & Picton, 2017*) | *Zelentia ninel* *Korshunova, Martynov & Picton, 2017* | ZMMU:Op-509 | Russia: Barents Sea | MF523400 | KY952178 | MF523242 | *Korshunova et al. (2017)* |
| *Tenellia ornata* (Baba, 1937) | *Cuthona ornata* Baba, 1937 | CASIZ 180344 | Hawaii | KY128758 | KY128967 | KY128553 | *Cella et al. (2016)* |

(Continued)

| Table 1 (continued) | | | | | | | |
|---|---|---|---|---|---|---|---|
| Present study species identifications | Previous species identifications | Voucher | Locality | Accession numbers 16S | Reference | | |
| | | | | | COI | H3 | |
| Tenellia osezakiensis (Martynov, Korshunova, Lundin & Malmberg, 2022) | Catriona osezakiensis Martynov, Korshunova, Lundin & Malmberg, 2022 | CASIZ 185133 | Hawaii | KY128701 | KY128909 | KY128496 | Cella et al. (2016) |
| Tenellia poritophages (Rudman, 1979) | Phestilla poritophages (Rudman, 1979) | CASIZ 177737 | Philippines | KY128759 | KY128968 | KY128554 | Cella et al. (2016) |
| Tenellia punicea (Millen, 1986) | Cuthonella punicea (Millen, 1986) | CASIZ 181525 | Canada | KY128761 | – | KY128556 | Cella et al. (2016) |
| Tenellia pustulata (Alder and Hancock, 1854) | Zelentia pustulata (Alder & Hancock, 1854) | WS3467 | Barents sea | KY128886 | KY129095 | KY128678 | Cella et al. (2016) |
| Tenellia puti Kim & Gosliner sp. nov. | Trinchesia yamasui (Hamatani, 1993) | CASIZ 177553 | Philippines | KY128796 | KY129004 | KY128591 | Cella et al. (2016) |
| Tenellia puti Kim & Gosliner sp. nov. | Trinchesia yamasui (Hamatani, 1993) | CASIZ 177554 | Philippines | KY128797 | KY129005 | KY128592 | Cella et al. (2016) |
| Tenellia puti Kim & Gosliner sp. nov. | Trinchesia yamasui (Hamatani, 1993) | CASIZ 177469 | Philippines | KY128794 | KY129002 | KY128589 | Cella et al. (2016) |
| Tenellia puti Kim & Gosliner sp. nov. | | CASIZ 186220 | Philippines | PP759733 | PP747812 | PP750952 | Present study |
| Tenellia puti Kim & Gosliner sp. nov. | | NMP 041346 / CASIZ 217192 | Philippines | PP759731 | PP747810 | PP750950 | Present study |
| Tenellia roginskae (Korshunova et al., 2018b) | Zelentia roginskae Korshunova et al., 2018b | ZMMU:Op-625 | Russia: White Sea | MH614974 | MH614983 | MH614994 | Korshunova et al. (2018b) |
| Tenellia sibogae (Bergh, 1905) | Phestilla sibogae Bergh, 1905 | CASIZ 177489 | Philippines | KY128767 | KY128975 | KY128562 | Cella et al. (2016) |
| Tenellia spadix (MacFarland, 1966) | Catriona spadix (MacFarland, 1966) | CASIZ 185195 | California | KY128698 | KY128906 | KY128493 | Cella et al. (2016) |
| Tenellia speciosa (Macnae, 1954) | Cuthona speciosa (Macnae, 1954) | CASIZ 176185 | South Africa | KY128787 | KY128995 | KY128582 | Cella et al. (2016) |
| Tenellia speciosa (Macnae, 1954) | Cuthona speciosa (Macnae, 1954) | CASIZ 176914 | South Africa | KY128789 | KY128997 | KY128584 | Cella et al. (2016) |
| Tenellia stipata (Alder & Hancock, 1843) | Diaphoreolis stipata (Alder & Hancock, 1843) | ZMMU:Op-376 | Russia: Barents Sea | OQ779047 | OQ779521 | OQ787052 | Korshunova et al. (2023a) |
| Tenellia subodiosa (Wang et al., 2020) | Phestilla subodiosa Wang et al., 2020 | PS3 | Thailand | MN255476 | MN255478 | MN255484 | Wang et al. (2020) |
| Tenellia viei (Mehrotra et al., 2020) | Phestilla viei Mehrotra et al., 2020 | PV2 | Thailand | MN239113 | MN257607 | MN257609 | Mehrotra et al. (2020) |
| Tenellia viridis (Forbes, 1840) | Diaphoreolis viridis (Forbes, 1840) | GnM9103 | Great Britain | KY128818 | KY129027 | KY128613 | Cella et al. (2016) |
| Tenellia willowsi (Korshunova et al., 2018b) | Zelentia willowsi Korshunova et al., 2018b | ZMMU:Op-629 | USA: Washington | MH614978 | MH614987 | MH614998 | Korshunova et al. (2018b), |

| Present study species identifications | Previous species identifications | Voucher | Locality | Accession numbers 16S | Reference COI | H3 | |
|---|---|---|---|---|---|---|---|
| *Tenellia yamasui* Hamatani, 1993 | *Trinchesia yamasui* (Hamatani, 1993) | CASIZ 182828 | Philippines | PP751617 | PP751619 | PP768732 | Present study |
| *Tenellia zvezda* (Korshunova et al., 2023a) | *Diaphoreolis zvezda* Korshunova et al., 2023a | ZMMU:Op-832 | Russia | OQ779049 | OQ779524 | OQ787063 | Korshunova et al. (2023a) |
| *Tenellia* sp. 6 | *Cuthona* sp. 2 | CASIZ 177293 | Philippines | KY128777 | KY128985 | KY128572 | Cella et al. (2016) |
| *Tenellia* sp. 13 | *Cuthona* sp. D | CASIZ 176796 | Hawaii | KY128771 | KY128979 | KY128566 | Cella et al. (2016) |
| *Tenellia* sp. 27 | *Cuthona* sp. 17 | CASIZ 177725 | Philippines | KY128773 | KY128981 | KY128568 | Cella et al. (2016) |
| *Tenellia* sp. 29 | *Cuthona* sp. 29 | CASIZ 180395 | Philippines | KY128776 | KY128984 | KY128571 | Cella et al. (2016) |
| *Tenellia* sp. 39 | *Cuthona* sp. 15 | CASIZ 181254 | Philippines | KY128772 | KY128980 | KY128567 | Cella et al. (2016) |
| *Tenellia* sp. 46 | *Cuthona* sp. 12 | CASIZ 176733 | Malaysia | KY128770 | KY128978 | KY128565 | Cella et al. (2016) |
| *Tenellia* sp. 46 | *Cuthona* sp. 19 | CASIZ 177316 | Philippines | KY128774 | KY128982 | KY128569 | Cella et al. (2016) |
| *Tenellia* sp. 46 | *Cuthona* sp. 19 | CASIZ 177722 | Philippines | KY128775 | KY128983 | KY128570 | Cella et al. (2016) |
| *Tenellia* sp. 52 | *Cuthona* sp. 10 | CASIZ 177583 | Philippines | KY128769 | KY128977 | KY128564 | Cella et al. (2016) |
| *Tenellia* sp. 58 | *Cuthona* sp. A | CASIZ 177747 | Philippines | KY128784 | KY128992 | KY128579 | Cella et al. (2016) |
| *Tenellia* sp. 59 | *Cuthona* sp. B | CASIZ 180404 | Philippines | KY128785 | KY128993 | KY128580 | Cella et al. (2016) |
| *Tenellia* sp. 59 | *Cuthona* sp. B | CASIZ 181244 | Philippines | KY128786 | KY128994 | KY128581 | Cella et al. (2016) |
| *Tenellia* sp. 79 | *Phestilla* sp. 3 | CASIZ 177518 | Philippines | KY128871 | KY129080 | KY128665 | Cella et al. (2016) |
| *Tenellia* sp. A | *Phestilla* sp. 1 | AF-2006 | Palau | DQ417272 | DQ417324 | – | Faucci, Toonen & Hadfield (2007) |
| *Tenellia* sp. B | *Phestilla* sp. 2 | AF-2006 | Guam | DQ417239 | DQ417285 | – | Faucci, Toonen & Hadfield (2007) |
| *Tenellia* sp. South Africa | *Cuthona* sp. C | CASIZ 176952 | South Africa | KY128718 | KY128923 | KY128511 | Cella et al. (2016) |
| *Tergipes tergipes* (Forsskål in Niebuhr, 1775) | | CASIZ 182699 | Maine | KY128878 | KY129087 | KY128670 | Cella et al. (2016) |
| *Tergipes tergipes* (Forsskål in Niebuhr, 1775) | | MNCN 15.05/67225 | Italy | KJ434063 | KJ434076 | KJ434094 | Cella et al. (2016) |

(Continued)

| Table 1 (continued) | | | | | | | |
|---|---|---|---|---|---|---|---|
| Present study species identifications | Previous species identifications | Voucher | Locality | Accession numbers 16S | Reference | | |
| | | | | | COI | H3 | |
| *Tergipes tergipes* (Forsskål in Niebuhr, 1775) | | MNCN 15.05/ 67228 | Netherlands | KJ434057 | KJ434072 | KJ434088 | *Cella et al. (2016)* |
| *Tergipes tergipes* (Forsskål in Niebuhr, 1775) | | MNCN 15.05/ 67235 | Wales | KJ434052 | KJ434067 | KJ434082 | *Cella et al. (2016)* |
| *Tergiposacca longicerata Cella et al., 2016* | *Tergipes* sp. | CASIZ 177605 | Philippines | KY128877 | KY129086 | KY128669 | *Cella et al. (2016)* |

94 °C, annealing for 1 min at 50–55 °C, and extension for 2 min at 72 °C with a final extension period of 5 min at 72 °C. Successful PCR products were visualized using electrophoresis on 1% agarose gel and then cleaned following the standard protocol for ExoSap-IT™ (USB, Affymetrix, Fremont, CA). The clean PCR products were fluorescently labeled with dye-terminators (Big Dye 3.1; Applied Biosystems, Foster City, CA, USA) during cycle sequencing following SteP protocol (*Platt, Woodhall & George, 2007*). Each reaction contained: 1.5 µL of 5× reaction buffer, 0.3 µL of primer (10 mM stock), 0.75 µL of Big Dye, 4.45–5.45 µL of Millipore-H2O, and 2–3 µL of cleaned PCR product. The newly labeled single-stranded DNA was precipitated using EDTA and then sequentially washed in 100% and 70% ethanol. HiDi formamide (10 µL, Applied BioSystems, Foster City, CA, USA) was added to each DNA pellet, denatured 95 °C for 2 min, and then immediately cooled on ice for 5 min. Both directions of the DNA fragments were sequenced on an ABI3130 Genetic Analyzer in the Center for Comparative Genomics at the California Academy of Sciences.

## Sequence alignment and analyses

For each gene fragment, both strands sequenced were assembled, trimmed to remove primers, and manually edited using Geneious v9.0 (*Kearse et al., 2012*). Single gene datasets were aligned with MAFFT (*Katoh, Asimenos & Toh, 2009*) using the algorithm E-INS-I with additional editing by hand for the 16S rRNA alignment. Each gene was analyzed independently using Bayesian inference (BI) and maximum likelihood (ML) and then concatenated into a three-gene alignment (16S rRNA+COI+H3). For the BI analysis, the best-fit evolution model and partitions were determined using ModelTest-NG (*Darriba et al., 2020*). Bayesian inference analysis was performed in MrBayes v3.2.6 (*Ronquist & Huelsenbeck, 2003*) with the concatenated dataset partitioned by gene and codon position. The following evolution models were used: GTR+GAMMA+I (16S rRNA, COI codon position 1, and H3 codon position 2), GTR+GAMMA (COI codon position 3 and H3 codon position 1), HKY+GAMMA+I (COI codon position 2), and K80 (H3 codon position 3). The BI analysis was run for $5.0 \times 10^7$ generations with Markov chains sampled

every 1,000 generations. Convergence of the two chains was checked using TRACER (*Drummond & Rambaut, 2007*) and then the initial 25% of estimated trees were removed as burn-in. The remaining tree estimates were used to create a 50% majority rule consensus tree with calculated posterior probabilities (pp). Non-parametric bootstrap values (bs) were estimated using randomized accelerated maximum likelihood (RAxML) v8.2.7 (*Stamatakis, 2014*). Each gene and codon position in the ML analysis used the evolution model GTR+GAMMA+I and was run for $5 \times 10^4$ bootstraps runs. Branches with posterior probabilities ≥0.95 and bootstrap values ≥70 were considered strongly supported, while posterior probabilities ≤0.95 and bootstrap values ≤70 were considered weakly supported (*Alfaro, Zoller & Lutzoni, 2003*).

## Species delimitation analyses

Species were only delimited in the three clades with the newly described species from this study to provide genetic comparisons with sister species and those with similar morphology. Three different approaches were implemented (1) assemble species by automatic partitioning (ASAP) method by *Puillandre, Brouillet & Achaz (2021)*, (2) the Bayesian Poisson tree process (bPTP) by *Zhang et al. (2013)*, and (3) the general mixed Yule coalescent (GMYC) model approach (*Pons et al., 2006*, *Fujisawa & Barraclough, 2013*). The ASAP method detects intraspecific and interspecific variation using genetic pairwise distances and then creates a scoring system to determine the best partition without relying on *a priori* species hypotheses. An ingroup COI alignment and an ingroup 16S rRNA alignment created in Geneious were uploaded to the ASAP Web-based interface (https://bioinfo.mnhn.fr/abi/public/asap/asapweb.html). Bayesian PTP models the number of substitutions between branches of previously inputted phylogenetic tree branches and identifies groups descended from a single ancestor using Bayesian MCMC methods. This test was performed separately on each of the three clades using a pruned 16S rRNA+COI+H3 concatenated BI tree on the bPTP server (https://species.h-its.org/) with the following parameters: 500,000 generations, 100 thinning, 0.1 burn-in, and 123 seeds. The GMYC approach assumes that different branching rates due to species diversification and independent evolution, which can then be delimited using a Yule Model (*Fujisawa & Barraclough, 2013*). Individual COI and 16S rRNA alignments were used as inputs to estimate ultrametric trees in BEAST v1.10.4 using the following priors previously assigned in BEAUTI v1.10.4: uncorrelated lognormal relaxed clock, yule speciation process, GTR +GAMMA+I, and 10 million generations MCMC chain length sampled every 1,000 steps. The first 10% of trees were removed for burn-in and a maximum clade credibility tree was estimated from the remaining tree estimates using TreeAnnotator v1.10.4. The GMYC modeling approach was implemented using the "single threshold" model (*Pons et al., 2006*) in the package Species Limits by Threshold Statistics (SPLITS, v1.0-20, *Fujisawa & Barraclough, 2013*) in R v4.2.1. Uncorrected pairwise genetic distances (*p*-distances) for COI and 16S rRNA were generated with Jukes-Cantor (JC69) settings in MEGA 11 v11.0.13 (*Tamura, Stecher & Kumar, 2021*).

## Morphological study

Specimens of new species described in this study were collected from the Philippines and Papua New Guinea *via* scuba diving. All the specimens from the Philippines were collected under our Gratuitous Permits (GP-0077-14, GP-0085-15) from the shallow waters of the municipalities of Mabini, Tingloy, Calatagan and Puerto Galera. Features of living animals were recorded in the field and from photographs. All the specimens were preserved in either Bouin's fixative, 75% or 95% EtOH. Morphological analyses were undertaken using a Nikon SMZ-U dissecting scope to determine the overall morphology of external structures, radula, jaws and details of the arrangement of reproductive organs. The penis was also examined for each specimen. In all instances the buccal mass was dissolved in a 10% solution of sodium hydroxide (NaOH) for approximately 24 h and then the elements of the buccal armature were rinsed in deionized water and prepared on glass coverslips for microscopic examination. Elements of the digestive and details of the reproductive systems were drawn using a camera lucida drawing attachment on dissecting scope. The penis was dissected from the reproductive system and mounted on glass coverslips for imaging. The SEM samples were mounted onto SEM stubs and then covered with gold/palladium using a Cressington 108 Auto vacuum sputter coater. Imaging of the jaws, radular teeth, and penis was done on a Hitachi SU3500 scanning electron microscope. Cerata were examined for the presence of nematocysts in the cnidosacs for some species by placing the tip of a ceras on a microscope slide and examining it using a compound light microscope. External features were examined directly, by photographs, or by literature review depending on the availability of specimens. Specimens and dissected structures were deposited at the California Academy of Sciences Department of Invertebrate Zoology collection (CASIZ) and the National Museum of the Philippines (NMP).

## Nomenclature acts

The electronic version of this article in Portable Document Format (PDF) will represent a published work according to the International Commission on Zoological Nomenclature (ICZN), and hence the new names contained in the electronic version are effectively published under that Code from the electronic edition alone. This published work and the nomenclatural acts it contains have been registered in ZooBank, the online registration system for the ICZN. The ZooBank LSIDs (Life Science Identifiers) can be resolved and the associated information can be viewed through any standard web browser by appending the LSID to the prefix http://zoobank.org/. The LSID for this publication is urn:lsid:zoobank.org:pub:1757F835-C668-4300-961A-FC1E8B409CE7. The online version of this work is archived and available from the following digital repositories: PeerJ, PubMed Central SCIE and CLOCKSS.

# RESULTS

## Phylogenetic relationships

The final concatenated dataset was 1,448 bp in length including gaps; however, not all three genes were successfully sequenced for all specimens used in the analyses (Table 1). The BI and ML analyses (Fig. 1, Figs. S1, S2) of the three-gene concatenated dataset

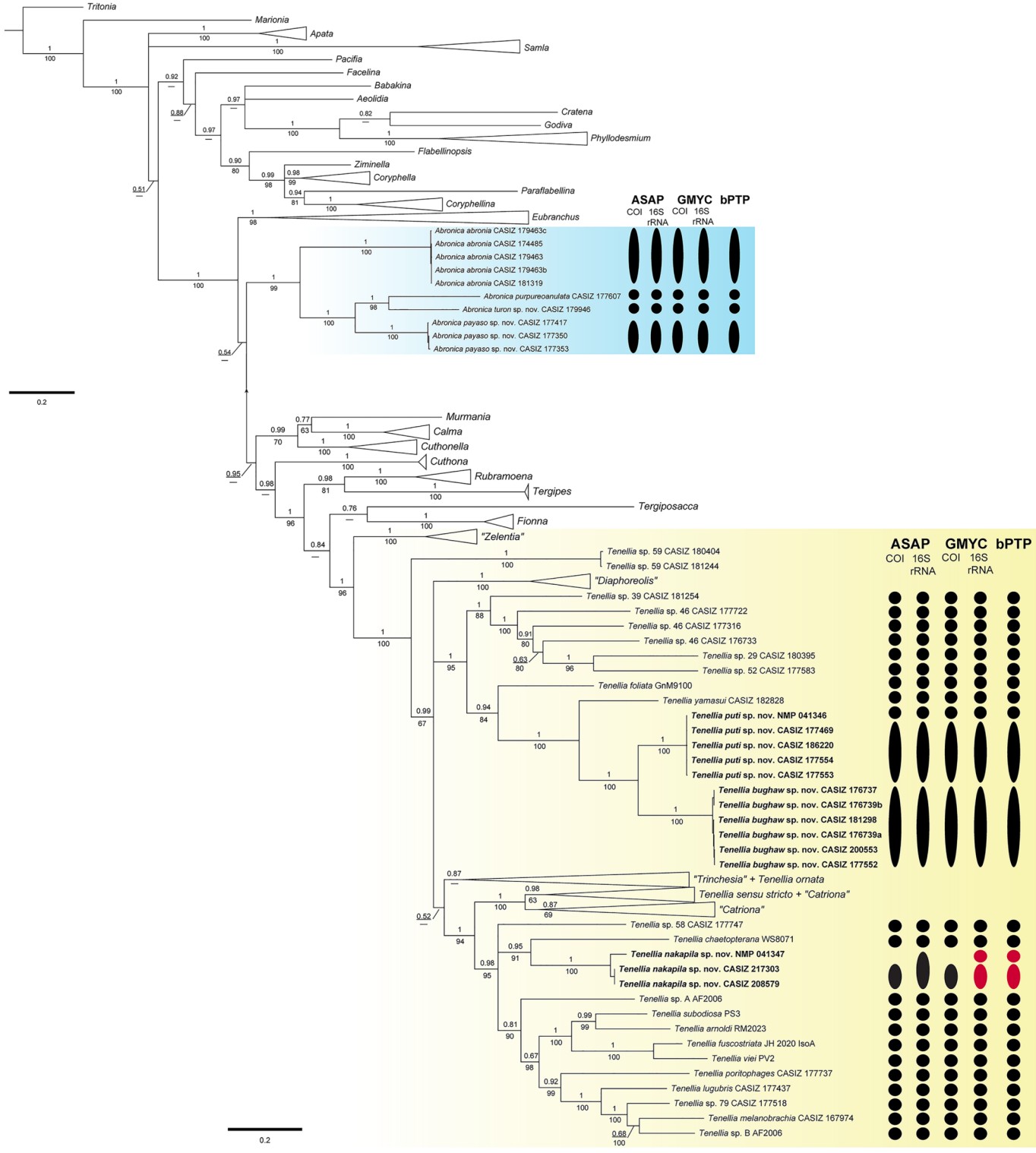

**Figure 1 Bayesian phylogenetic tree of various aeolid nudibranchs estimated from the three gene (16S rRNA+COI+H3) concatenated data.** Numbers above branches refer to BI posterior probabilities (pp), while numbers below branches refer to ML non-parametric bootstrapping values (bs). Dashes indicate relationships not recovered during the ML analysis. Subclades with new species are highlighted with color. Results of the species delimitation analyses: ASAP, GMYC, and bPTP are to the right and colored based on partitioning: black = a single species and red = over-partitioning.

resulted in a similarly supported topology as Cella et al. (2016). In the genus *Abronica* (highlighted in light blue in Fig. 1), *Abronica abronia* (MacFarland, 1966) is basal in a well-supported (pp = 1.00, bs = 99) clade that includes the newly described *Abronica payaso* sp. nov. and *Abronica turon* sp. nov. as well as *Abronica purpureoanulata* (Baba, 1961). The genus *Tenellia* (highlighted in yellow in Fig. 1) is mostly well-supported (pp = 100, bs = 96) and includes species previously assigned to *Zelentia*, *Trinchesia*, *Diaphoreolis*, *Catriona*, and *Phestilla*.

In *Tenellia* the lowest nodes of the clade are well-supported (pp = 1, bs = 100) and composed of species previously identified as "*Zelentia*" and an undescribed species of *Tenellia*. The rest of *Tenellia* is composed of a polytomy between three large clades. The first clade is a well-supported clade (pp = 1, bs = 100) of species previously identified as "*Diaphoreolis*". The second clade is a well-supported clade (pp = 1, bs = 95) composed of mostly undescribed species from the Central Pacific. It includes a specimen of *Tenellia foliata* (Forbes & Goodsir, 1839) and *T. yamasui*, which were previously assigned to "*Trinchesia*", the newly described *Tenellia puti* sp. nov. and its sister species *Tenellia bughaw* sp. nov., and the undescribed *Tenellia* sp. 29, *Tenellia* sp. 39, *Tenellia* sp. 46, and *Tenellia* sp. 52.

The third clade of *Tenellia* is un-supported (pp = 0.52, bs = NA) and includes a weakly-supported subclade (pp = 0.87, bs = NA) of the rest of "*Trinchesia*" and a specimen of *Tenellia ornata* (Baba, 1937) which was also included in "*Trinchesia*". The second subclade is well-supported (pp = 1, bs = 100) and is composed of *Tenellia* sensu *stricto*, *Tenellia* sp. 13, and "*Catriona*" *gymnota*, and "*Catriona*" *aurantia*, and a clade of the rest of "*Catriona*". The third subclade is well-supported (pp = 0.98, bs = 95) and has a polytomy with undescribed species of *Tenellia* and species previously assigned to "*Phestilla*". This polytomy is composed of a single specimen of the undescribed *Tenellia* sp. 58, a well-supported clade (pp = 0.95, bs = 91) of a single specimen of *Tenellia chaetopterana* with the newly described *T. nakapila* sp. nov., and a larger less supported (pp = 0.81, bs = 90) clade of *Tenellia lugubris* (Bergh, 1870), *Tenellia melanobranchia* (Bergh, 1874), *Tenellia poritophages* (Rudman, 1979), *Tenellia subodiosa* (Wang et al., 2020); *Tenellia arnoldi* (Mehrotra & Caballer, 2024), *Tenellia fuscostriata* (Hu et al., 2020), *Tenellia viei* (Mehrotra et al., 2020), and the undescribed *Tenellia* sp. A from Palau and *Tenellia* sp. B from Guam.

## Species delimitation

The COI ASAP analysis (Table 2, Fig. S3) and 16S rRNA ASAP analysis (Table 3, Fig. S4) using Jukes-Cantor (JC69) both recovered four partitions within the genus *Abronica* (ASAP scores of 2.00 and 1.00, respectively) and ten partitions within the *Tenellia* clade including *T. foliata*, *T. yamasui* and the newly described *T. puti* sp. nov. and *T. bughaw* sp. nov. (ASAP score 1.50 for both analyses). The COI ASAP analysis recovered eleven partitions and the 16S rRNA ASAP analysis recovered fourteen partitions within the *Tenellia* clade including *T. poritophages*, *T. melanobranchia*, *T. subodiosa*, *T. arnoldi*, *T. fuscostriata*, *T. viei*, *T. chaetopterana* and the newly described *T. nakapila* sp. nov. (ASAP scores of 3.00 and 2.00, respectively). The COI ASAP analysis with eleven

Table 2 **Pairwise distances.** Pairwise uncorrected p-distances (%) for COI between *Abronica* spp. and a subset of *Tenellia* spp. with intraspecific p-distances in bold.

|   |   | 1 | 2 | 3 | 4 |
|---|---|---|---|---|---|
| 1 | *Abronica abronia* | **0.0–0.8** | | | |
| 2 | *Abronica purpureoanulata* | 23.6 | – | | |
| 3 | *Abronica turon* sp. nov. | 20.9 | 20.7 | – | |
| 4 | *Abronica payaso* | 21.6 | 20.5 | 19.9 | **0.0–0.3** |

|   |   | 1 | 2 | 3 | 4 | 5 | 6 | 7 | 8 | 9 | 10 |
|---|---|---|---|---|---|---|---|---|---|---|---|
| 1 | *Tenellia foliata* | – | | | | | | | | | |
| 2 | *Tenellia* sp. 46 CASIZ 176733 | 20.2 | – | | | | | | | | |
| 3 | *Tenellia* sp. 39 | 20.6 | 18.5 | – | | | | | | | |
| 4 | *Tenellia* sp. 46 CASIZ 177316 | 21.2 | 16.2 | 18.5 | – | | | | | | |
| 5 | *Tenellia* sp. 46 CASIZ 177722 | 20.8 | 20.7 | 22.6 | 19.5 | – | | | | | |
| 6 | *Tenellia* sp. 29 | 20.1 | 19.1 | 20.5 | 18.9 | 22.2 | – | | | | |
| 7 | *Tenellia* sp. 52 | 21.4 | 16.6 | 19.7 | 19.5 | 18.9 | 18.5 | – | | | |
| 8 | *Tenellia yamasui* | 20.4 | 22.2 | 20.9 | 21.1 | 22.1 | 20.1 | 18.7 | – | | |
| 9 | *Tenellia bughaw* sp. nov. | 21.6 | 18.5 | 21 | 18 | 21.8 | 20 | 19.1 | 20.6 | **0.0–0.9** | |
| 10 | *Tenellia puti* sp. nov. | 21.8 | 19.3 | 22.4 | 21.3 | 19.9 | 22.4 | 20.9 | 18.1 | 14.1 | **0.2–0.6** |

|   |   | 1 | 2 | 3 | 4 | 5 | 6 | 7 | 8 | 9 | 10 | 11 | 12 | 13 |
|---|---|---|---|---|---|---|---|---|---|---|---|---|---|---|
| 1 | *Tenellia arnoldi* | – | | | | | | | | | | | | |
| 2 | *Tenellia subodiosa* | 11.6 | – | | | | | | | | | | | |
| 3 | *Tenellia fuscostriata* | 14.2 | 12.9 | – | | | | | | | | | | |
| 4 | *Tenellia viei* | 15.5 | 16.1 | 12.2 | – | | | | | | | | | |
| 5 | *Tenellia* sp. A | 15.3 | 14.5 | 14.9 | 17.0 | – | | | | | | | | |
| 6 | *Tenellia* sp. B | 17.0 | 14.5 | 15.7 | 17.6 | 16.8 | – | | | | | | | |
| 7 | *Tenellia* sp. 79 | 17.9 | 16.1 | 18.2 | 18.2 | 16.6 | 14.5 | – | | | | | | |
| 8 | *Tenellia melanobranchia* | 18.3 | 16.9 | 16.6 | 17.4 | 18.7 | 14.7 | 16.6 | – | | | | | |
| 9 | *Tenellia chaetopterana* | 18.0 | 16.2 | 17.3 | 19.4 | 15.9 | 18.5 | 18.2 | 22.0 | – | | | | |
| 10 | *Tenellia nakapila* sp. nov | 18.1 | 17.5 | 17.8 | 18.9 | 18.3 | 17.8 | 19.5 | 19.1 | 17.5 | 0.0 | | | |
| 11 | *Tenellia poritophages* | 17.7 | 16.3 | 17.2 | 19.3 | 15.7 | 19.1 | 18.4 | 20.8 | 17.3 | 16.4 | – | | |
| 12 | *Tenellia lugubris* | 20.3 | 18.8 | 20.3 | 22.2 | 21.1 | 19.4 | 18.9 | 22.0 | 20.8 | 20.6 | 21.0 | – | |
| 13 | *Tenellia* sp. 58 | 23.4 | 20.7 | 20.7 | 20.0 | 20.0 | 19.8 | 21.8 | 20.4 | 22.0 | 20.4 | 21.0 | 22.4 | – |

partitions fails to split *T. arnoldi* from *T. subodiosa* and *T. fuscostriata* from *T. viei*. All four species are morphologically distinct, therefore the partition with thirteen groups is congruent with our molecular and morphological analyses (ASAP score of 6.00). The 16S rRNA ASAP analysis with fourteen partitions over splits one specimen of *T. nakapila* sp. nov.; however, all three specimens are from the Philippines and are morphologically identical. Therefore, the partition with thirteen groups (ASAP score = 8.00) is congruent with our other analyses.

The three-gene bPTP analysis recovered the same four partitions within *Abronica and* the same ten partitions in the first *Tenellia* subclade; however, the second *Tenellia* subclade was over-partitioned. In the second subclade, fourteen partitions were recovered, but one

**Table 3 Pairwise distances.** Pairwise uncorrected p-distances (%) for 16S rRNA between *Abronica* spp. and a subset of *Tenellia* spp. with intraspecific p-distances in bold.

| | | 1 | 2 | 3 | 4 | | | | | | | | |
|---|---|---|---|---|---|---|---|---|---|---|---|---|---|
| 1 | *Abronica abronia* | **0.0–0.2** | | | | | | | | | | | |
| 2 | *Abronica purpureoanulata* | 15.1 | – | | | | | | | | | | |
| 3 | *Abronica turon* sp. nov. | 15.2 | 9.2 | – | | | | | | | | | |
| 4 | *Abronica payaso* sp. nov. | 16.1 | 10.7 | 11.4 | **0.2–1.1** | | | | | | | | |

| | | 1 | 2 | 3 | 4 | 5 | 6 | 7 | 8 | 9 | 10 | | |
|---|---|---|---|---|---|---|---|---|---|---|---|---|---|
| 1 | *Tenellia foliata* | – | | | | | | | | | | | |
| 2 | *Tenellia* sp. 46 CASIZ 176733 | 11.6 | – | | | | | | | | | | |
| 3 | *Tenellia* sp. 39 | 10.9 | 10.9 | – | | | | | | | | | |
| 4 | *Tenellia* sp. 46 CASIZ 177316 | 11.8 | 11.8 | 9.3 | – | | | | | | | | |
| 5 | *Tenellia* sp. 46 CASIZ 177722 | 10.9 | 10.9 | 10.8 | 12 | – | | | | | | | |
| 6 | *Tenellia* sp. 29 | 15.7 | 14.9 | 14.8 | 11.6 | 14.4 | – | | | | | | |
| 7 | *Tenellia* sp. 52 | 16.3 | 17.2 | 12.8 | 15.4 | 13.6 | 14.6 | – | | | | | |
| 8 | *Tenellia yamasui* | 12.7 | 18.1 | 13.6 | 19.2 | 15.8 | 18.9 | 20.1 | – | | | | |
| 9 | *Tenellia bughaw* sp. nov. | 12.7 | 17 | 13.6 | 15.2 | 17 | 19.2 | 19.8 | 11.9 | **0.0–0.5** | | | |
| 10 | *Tenellia puti* sp. nov. | 12.9 | 17.7 | 17.7 | 15.1 | 15 | 19.5 | 18.9 | 12.5 | 9.5 | **0.0–0.2** | | |

| | | 1 | 2 | 3 | 4 | 5 | 6 | 7 | 8 | 9 | 10 | 11 | 12 | 13 |
|---|---|---|---|---|---|---|---|---|---|---|---|---|---|---|
| 1 | *Tenellia arnoldi* | – | | | | | | | | | | | | |
| 2 | *Tenellia* sp. A | 12.8 | – | | | | | | | | | | | |
| 3 | *Tenellia* sp. B | 11.9 | 11.9 | – | | | | | | | | | | |
| 4 | *Tenellia melanobranchia* | 11.6 | 9.0 | 5.3 | – | | | | | | | | | |
| 5 | *Tenellia* sp. 79 | 12.4 | 11.0 | 5.3 | 4.2 | – | | | | | | | | |
| 6 | *Tenellia lugubris* | 13.0 | 10.1 | 7.5 | 6.9 | 6.2 | – | | | | | | | |
| 7 | *Tenellia subodiosa* | 8.2 | 10.4 | 9.3 | 8.0 | 7.7 | 8.7 | – | | | | | | |
| 8 | *Tenellia* sp. 58 | 10.5 | 10.1 | 9.6 | 8.5 | 9.5 | 11.3 | 8.4 | – | | | | | |
| 9 | *Tenellia poritophages* | 13.0 | 11.0 | 10.1 | 9.8 | 8.7 | 10.2 | 8.7 | 11.9 | – | | | | |
| 10 | *Tenellia nakapila* sp. nov. | 11.4 | 11.1 | 11.7 | 10.3 | 11.1 | 12.4 | 11.4 | 8.2 | 11.9 | **1.4** | | | |
| 11 | *Tenellia fuscostriata* | 12.3 | 13.2 | 12.0 | 11.1 | 11.7 | 12.0 | 10.8 | 11.4 | 11.7 | 14.1 | – | | |
| 12 | *Tenellia viei* | 11.6 | 14.4 | 11.7 | 11.1 | 11.6 | 13.5 | 11.1 | 12.1 | 11.9 | 12.2 | 5.2 | – | |
| 13 | *Tenellia chaetopterana* | 14.3 | 12.2 | 14.4 | 13.3 | 12.7 | 12.4 | 11.4 | 10.8 | 13.0 | 10.3 | 13.5 | 15.4 | – |

specimen of *T. nakapila* sp. nov. (NMP 041347) was split from the other two specimens potentially due to the lack of successful COI sequencing or perhaps due to high intraspecific divergence of the 16S rRNA alignment. The GMYC analysis for COI recovered the same number of partitions as the COI and 16S rRNA ASAP analyses; since, the 16S rRNA GMYC analysis also over-partitioned the *T. nakapila* sp. nov. clade into fourteen partitions similar to the bPTP analysis.

The maximum genetic distance for COI within the *Abronica* clade (highlighted in blue in Fig. 1) was 23.6% between *A. abronia* and *A. purpureoanulata*, while the maximum distance for 16S rRNA was 16.1% between *A. payaso* sp. nov. and *A. abronia*. Intraspecific variation within two species of *Abronica* ranged between 0.0–0.8% within COI and

0.0–1.1% within 16S rRNA; however, no intraspecific variation was calculated for *A. turon* sp. nov. due to a lack of additional sequenced specimens. Within the first subclade of *Tenellia* the maximum genetic distance for COI was 22.6% between *Tenellia* sp. 39 and a specimen of *Tenellia* sp. 46 (CASIZ 177722), while the maximum distance for 16S rRNA was similar (20.1%) between *Tenellia* sp. 52 and *T. yamasui*. Intraspecific variation for two species within this clade (*T. bughaw* sp. nov., and *T. puti* sp. nov.) ranged between 0.0–0.9% for COI and 0.0–0.5% for 16S rRNA. In the second subclade of *Tenellia* the maximum genetic distance for COI was 23.4% between *T. arnoldi* and *T. sp.* 58, while the maximum distance for 16S rRNA was only 14.4% between *T. viei* and *Tenellia* sp. A. Within this clade, intraspecific variation within one species, *T. nakapila* sp. nov., was 1.4% for 16S rRNA.

## Systematics

*Tenellia yamasui* (*Hamatani, 1993*)

Figures 2–4A

*Cuthona yamasui Hamatani, 1993*: 127, figs. 1-3.
*Cuthona* sp. 36 *Gosliner, Behrens & Valdés, 2008*: 370, misidentification

**Material examined**
CASIZ 182828, one specimen sequenced and dissected, ST 24, Bethlehem, 13.672800°N, 120.841339°E, Maricaban Island, Tingloy, Batangas Province, Luzon Philippines, 1 m depth, 20 May 2010, Alicia Hermosillo McKowen, collector.

**Distribution**
Originally described from Okinawa (*Hamatani, 1993*) and also recorded from Indonesia (*Gosliner, Behrens & Valdés, 2008*), and the Philippines (*Gosliner, Valdés & Behrens, 2015*).

**Natural history**
Found on shallow coral rubble habitats where it feeds on the thecate hydroid *Pachyrhynchia cupressina* (Kirchenpauer, 1872).

**External morphology**
The overall body color (Fig. 2) is an opaque white suffused with orange pigment. The concentration of orange pigment is particularly intense on the head and anterior portion of the body. Most of the body surface is covered with cerata. There is a thick honey-orange colored stripe traveling along the dorsal side of the body. The oral tentacles are opaque white in the basal half and bright orange in the outer half. Anterior to the oral tentacles is a bright orange spot on the front of the head. At the base of each oral tentacle, an opaque white line runs diagonally and unites into a medial longitudinal line that extends between the rhinophores and continues posteriorly a short distance behind the rhinophores. The rhinophore color is orange basally, followed by an opaque white band, another orange band, and a short opaque white apex. The thin, conical rhinophores are slightly longer

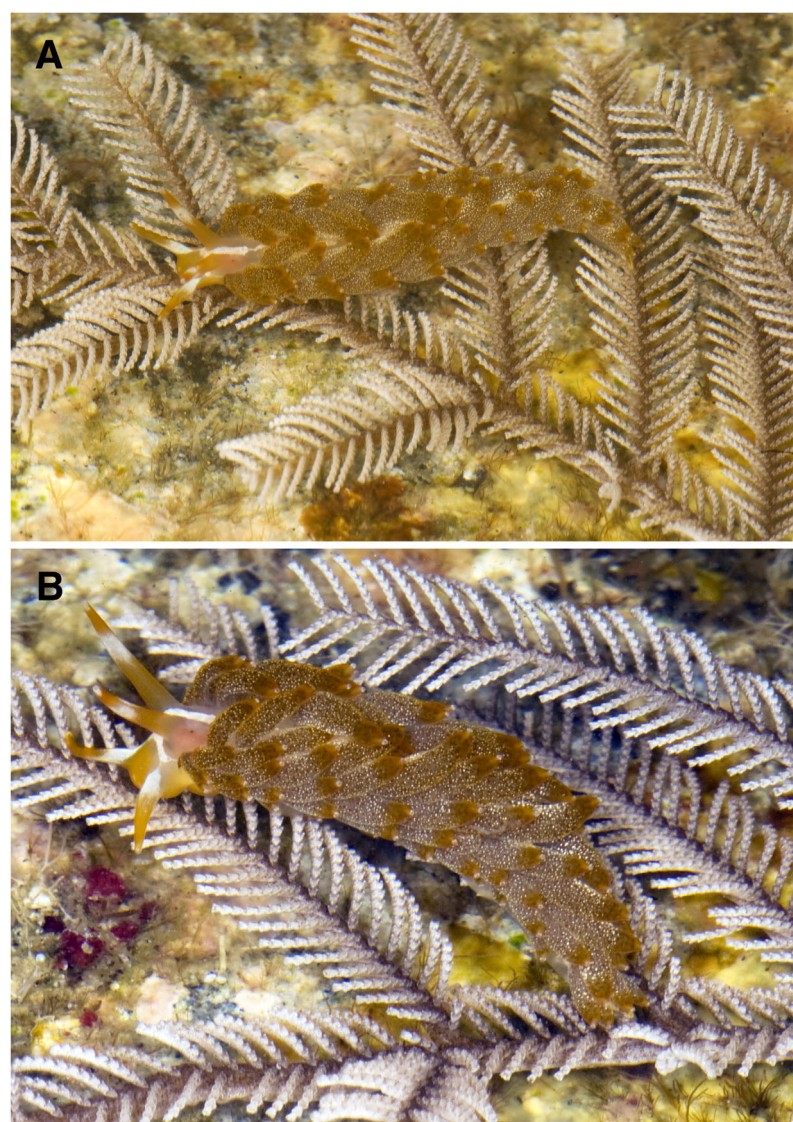

**Figure 2** ***Tenellia yamasui* living animals.** (A, B) Living animals of *Tenellia yamasui* (*Hamatani, 1993*), CASIZ 182828, on its prey hydroid *Pachyrhynchia cupressin*a, photos by T. Gosliner.

than the oral tentacles. The texture is smooth and they are narrowly tapered, resembling a perfect icicle.

The cerata are very specialized to camouflage on their prey hydroid. The overall color of the cerata is a rusty reddish-brown color. Numerous small opaque white spots are present on the cerata with the exception of the apical portion. The tips are a burnt orange color with a white color at the distal end where the cnidosacs are situated. The widest part of the cerata is in the middle. They are folded in a chevron pattern along the dorsal side of the body. The cerata are arranged in numerous linear rows. There are four rows in the precardiac ceratal rows. In the one specimen (CASIZ 182828) examined here, the

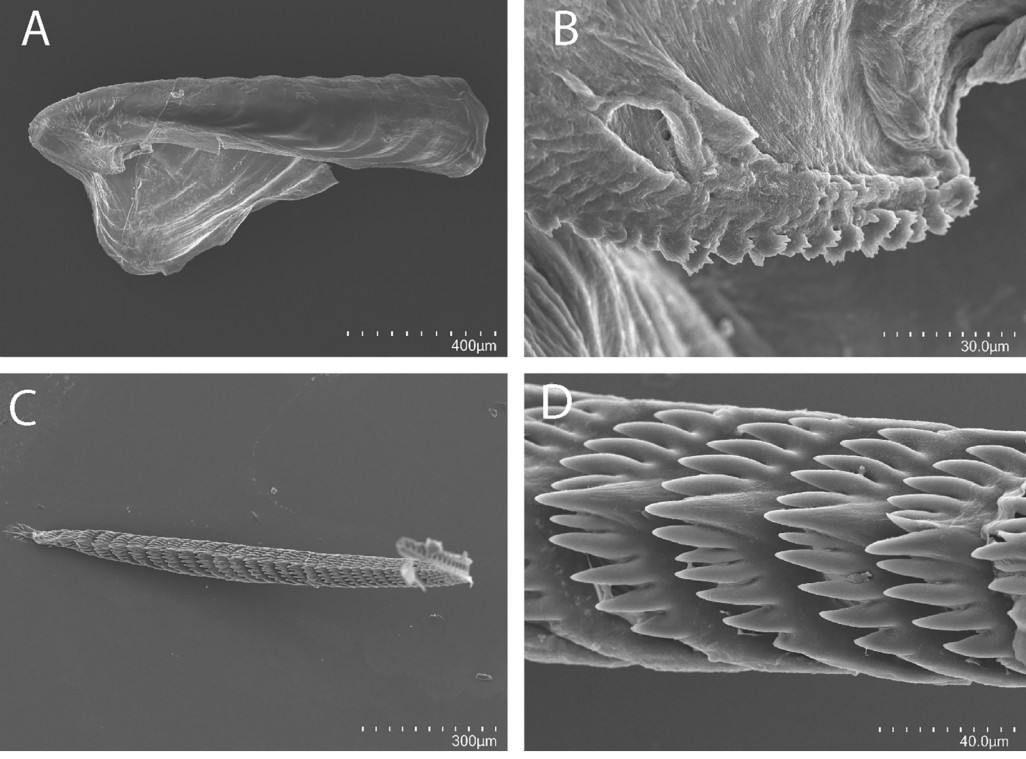

**Figure 3** *Tenellia yamasui* (*Hamatani, 1993*), **scanning electron micrographs of jaws and radula, CASIZ 182828.** (A) Entire jaw. (B) Masticatory border with individual denticles. (C) Entire radula. (D) Individual teeth.  

precardiac rows beginning with the most anterior row contain 4, 5, 4, 5 cerata per row. After the interhepatic space, there are 7–8 postcardiac ceratal rows, each of which contains 1–4 cerata. The anus is dorsal and acleioproctic and is located anterior to the first ceratal row of the postcardiac cerata. The genital opening is ventral to the third and fourth precardiac ceratal rows.

**Buccal mass and radula**

On either side of the buccal mass are large, dendritic oral glands that extend posteriorly two-thirds of the body length. In the one specimen examined (CASIZ 182828), the jaw resembles a clam shell shape. The texture has grooves like an undulating wave pattern. The jaws (Fig. 3A) are dark brown in color and relatively thickly cuticularized with a thickened area on the anterior side of the jaw. The masticatory margin is elongated and contains numerous irregular denticles (Fig. 3B) along its edge. The radula contains a single row of 54 teeth (Fig. 3C). The radula is elongated and relatively narrow with a few wider teeth near the middle of the ribbon. There is a prominent central denticle and the other denticles are positioned more posteriorly in an evenly graded fashion. There are 5–6 primary lateral cusps on either side of the central cusp (Fig. 3D). There are no smaller secondary denticles between the large denticles. The lateral cusps are acutely pointed and slightly curved inwardly.

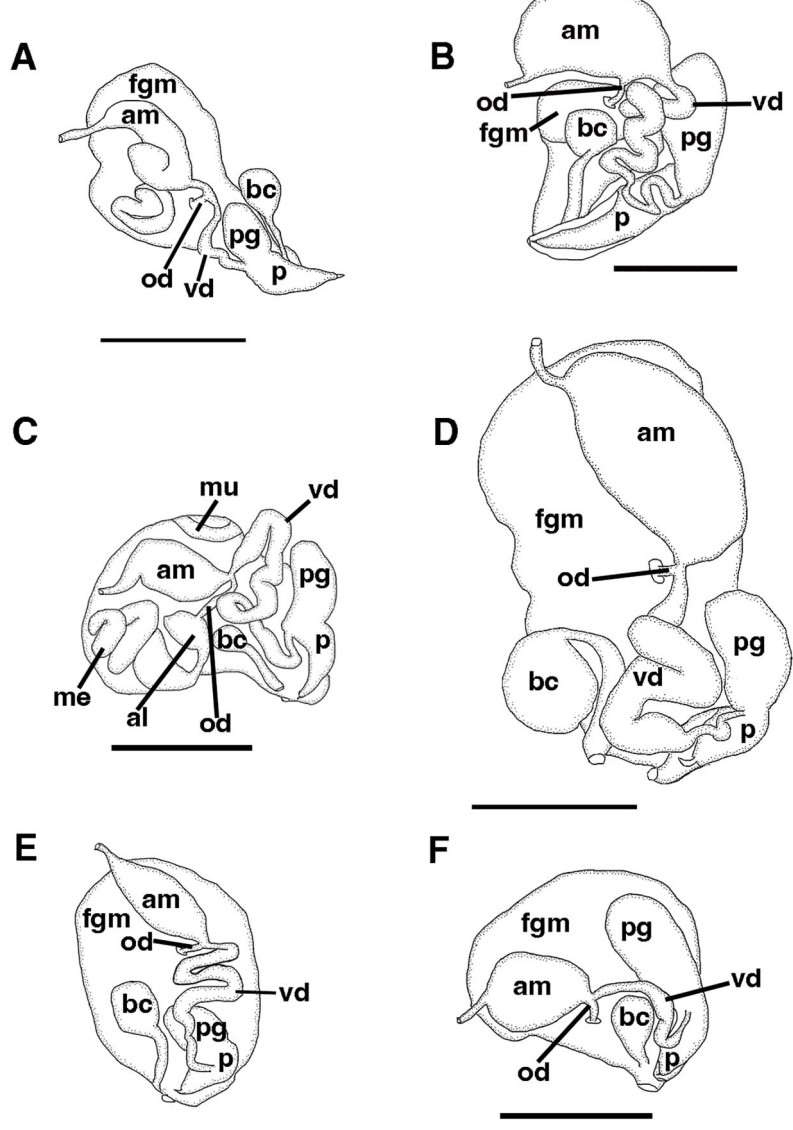

**Figure 4 Reproductive systems.** (A) *Tenellia yamasui* (*Hamatani, 1993*), CASIZ 182828, scale = 2.5 mm. (B) *Tenellia puti* Kim & Gosliner, sp. nov., CASIZ 186220, scale = 1.5 mm. (C) *Tenellia bughaw* Kim & Gosliner, sp. nov., CASIZ 200553, scale = 2.4 mm. (D), *Tenellia nakapila* Kim & Gosliner, sp. nov., CASIZ 217303, scale = 1.4 mm. (E) *Abronica payaso* Kim & Gosliner, sp. nov., CASIZ 241493, scale = 1.0 mm. (F) *Abronica turon* Kim & Gosliner, sp. nov., CASIZ 086449, scale = 0.9 mm., abbreviations: al-albumen gland, am-ampulla, bc-bursa copulatrix, fgm-female gland mass, me-membrane gland, mu-mucous gland, od-oviduct, p-penis, pg-penial gland, vd-vas deferens.

## Reproductive system

The reproductive system is androdiaulic (Fig. 4A). The ovotestis follicles contain a large female acinus surrounded by a series of smaller male acini. The large, saccate ampulla divides distally into the short oviduct and vas deferens. The vas deferens begins with a thick prostatic portion of and narrows into a short, convoluted ejaculatory duct, entering

the penis near the junction of the penial gland with the penial papilla. The penial gland is pyriform whereas the penial papilla is conical. The penial stylet is straight and has a conical shape (Fig. 4A). The female glands are well-developed and small albumen and membrane glands are clearly visible, as is the larger mucous gland. A spherical bursa copulatrix is present at the distal end of the reproductive system and connects to the gonopore *via* an elongated duct.

**Remarks**

The present specimen (CASIZ 182828) closely matches the original description of *Cuthona yamasui* by *Hamatani (1993)* from Okinawa, Japan, in its external morphology, color pattern, detail of the jaws and radula, and reproductive anatomy. The most distinguishing external features of this species are the presence of orange and white pigment on the head, rhinophores and oral tentacles and the rusty brown cerata with opaque white spots over their surface. The jaws and radular teeth closely resemble those described by Hamatani. Scanning electron microscopy reveals the presence of fine secondary denticles on the denticles of the masticatory margin of the present specimen. The radular teeth are characterized by having 5–6 evenly graduated lateral denticles on either side of the wider, triangular central cusp. The reproductive system also closely resembles that illustrated by Hamatani, but the bursa copulatrix of the present specimens has a more elongated duct. The penial stylet of our specimen was visible during dissection but was not evident in the scanning electron microscopic preparation.

Confusion around this species has arisen from considering species with very different color patterns as being color variants of the same species. Most of these variants have blue or green body pigment and lack the orange-white pigment on the head. *Rudman (2002)* and later postings on the Sea Slug Forum were identified as *C. yamasui*, but most of these variants represent the two species described subsequently as *T. puti* sp. nov. and *T. bughaw* sp. nov. This confusion was further exacerbated by *Gosliner, Behrens & Valdés (2008)*, where some of these color variants were identified as *C. yamasui* and the true *C. yamasui* was identified as *Cuthona* sp. 36. This was corrected in *Gosliner, Valdés & Behrens (2015)* where *C. yamasui* was correctly identified. In *Cella et al. (2016)* numerous species of *Cuthona*, *Catriona*, *Trinchesia*, and *Phestilla* were moved to the genus *Tenellia* including *C. yamasui*. Two additional similar-looking species depicted in *Gosliner, Valdés & Behrens (2015)* as *Cuthona* sp. 13 and *Cuthona* sp. 14, which correspond to *Tenellia* sp. E and *Tenellia* sp. F, respectively, in *Cella et al. (2016)*, are described as *T. puti* sp. nov. and *T. bughaw* sp. nov. in the present work. In the present study, we have shown that *T. yamasui* is a distinct species based on our phylogenetic analyses and is further corroborated by our morphological examination.

***Tenellia puti* Kim & Gosliner sp. nov**.

LSID:urn:lsid:zoobank.org:act:65A96380-78E1-48DC-B344-6D0FC8549A19

Figures 4B, 5A–5D, 6
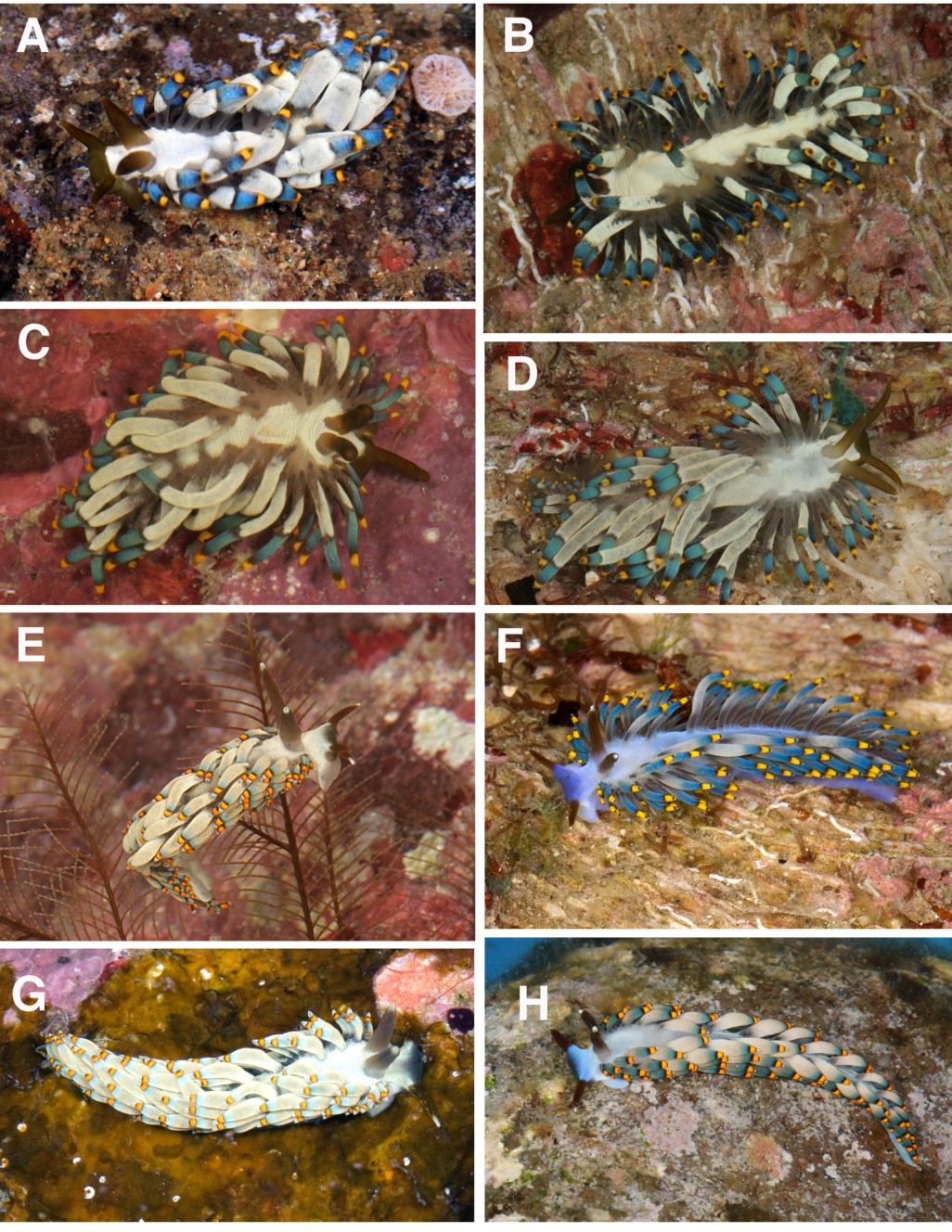

**Figure 5 Living animals, *Tenellia puti* and *T. bughaw*.** (A) *Tenellia puti* Kim & Gosliner sp. nov., paratype, CASIZ 186220, Mabini, Philippines. (B) *Tenellia puti* Kim & Gosliner sp. nov, CASIZ 177553, Mabini, Philippines. (C) *Tenellia puti* Kim & Gosliner sp. nov, CASIZ 177469, Mabini, Philippines. (D) *Tenellia puti* Kim & Gosliner sp. nov, CASIZ 177554, Mabini, Philippines. (E) *Tenellia bughaw* Kim & Gosliner sp. nov. CASIZ 176735, Pineapple Point, Pulau Tenggol, Malaysia. (F) *Tenellia bughaw* Kim & Gosliner sp. nov. CASIZ 177552, Mabini, Philippines. (G) *Tenellia bughaw* Kim & Gosliner sp. nov. CASIZ 176739, Pulau Varella, Malaysia. (H) *Tenellia bughaw* Kim & Gosliner sp. nov. CASIZ 200553, Tingloy, Philippines, all photos by T. Gosliner.

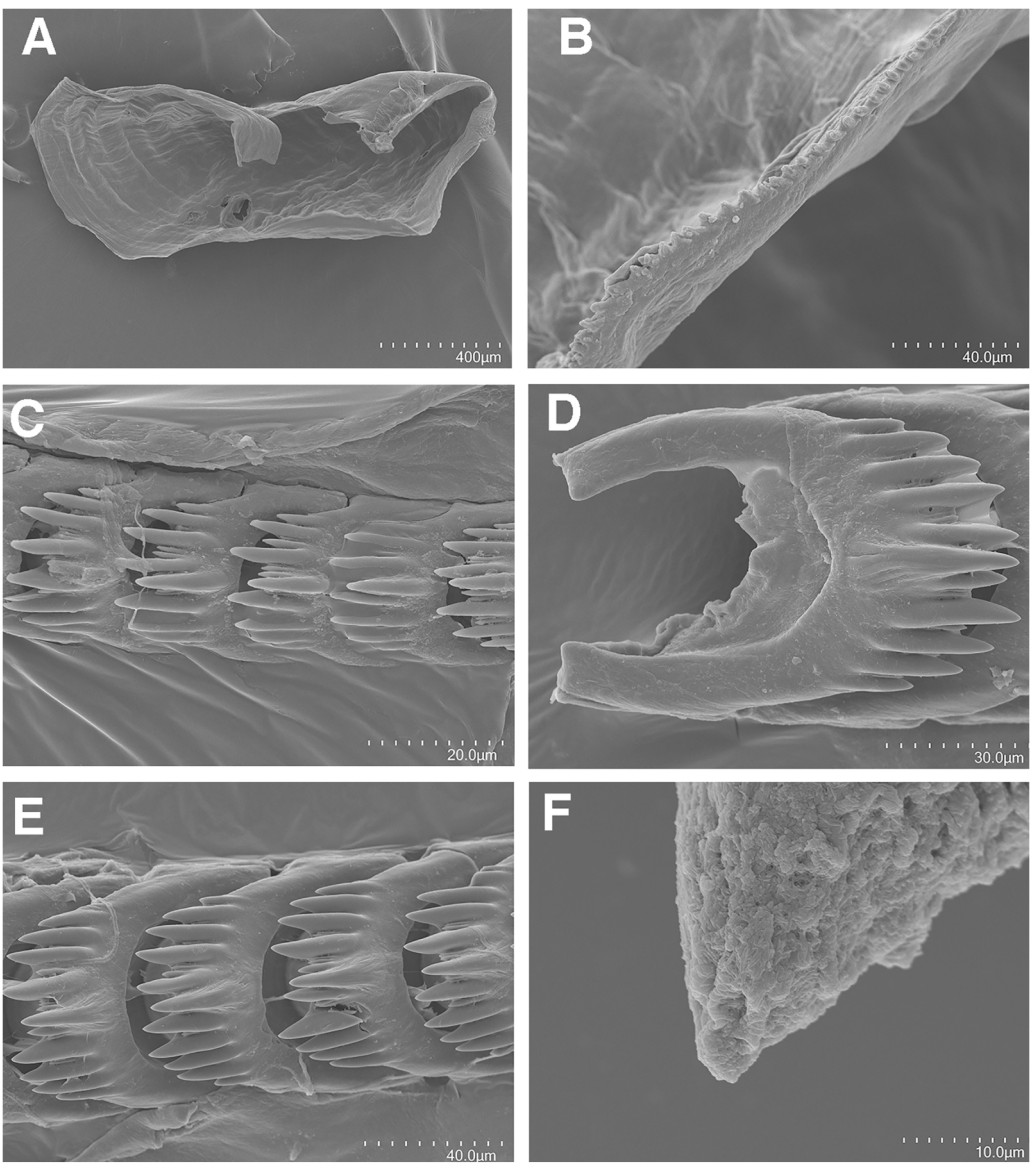

**Figure 6** *Tenellia puti* Kim & Gosliner sp. nov., scanning electron micrographs of jaws and radula, CASIZ 200553. (A) Entire jaw. (B) Masticatory border with individual denticles. (C) Entire radula. (D, E) Radular teeth. (F) Penial stylet.

*Cuthona yamasui Hamatani, 1993*: *Gosliner, Behrens & Valdés, 2008*: 360, first row, second row, middle and right photos, misidentification.

*Cuthona* sp. 13: *Gosliner, Valdés & Behrens, 2015*: 345, left photo.

*Tenellia* sp. E: *Cella et al., 2016*

*Tenellia* sp. 15: *Gosliner, Valdés & Behrens, 2018*: 287, left photo.

### Holotype

NMP 041346 (formerly CASIZ 217192) one specimen, sequenced, and dissected, ST DAU 06, 9.2007°N, 123.2789°E, San Miguel, Dauin, Negros Oriental, Philippines, 7 April, 2016, T. M. Gosliner.

**Paratypes**

CASIZ 177469, one specimen, sequenced and dissected, St 14, 13.6864°N, 120.8954°E, Mainit Point, Mabini Batangas, Luzon, Philippines, 20 March, 2008, T. Gosliner. CASIZ 177553, one specimen, sequenced and dissected, St 23, 13.6864°N, 120.8954°E, Mainit Point, Mabini Batangas, Luzon, Philippines, 21 March, 2008, T. Gosliner. CASIZ 177554, one specimen, sequenced and dissected, St 23, 13.6864°N, 120.8954°E, Mainit Point, Mabini Batangas, Luzon, Philippines, 21 March, 2008, T. Gosliner. CASIZ 186220, one specimen, St HEP 62, sequenced and dissected, St 14, 13.68627°N, 120.89544°E, Mainit Bubbles, Mabini Batangas, Luzon, Philippines, 13 May, 2011, T. Gosliner.

**Distribution**

This species is known from the Philippines (*Gosliner, Valdés & Behrens, 2015*, *2018*) and Indonesia (*Ianniello, 2003*; *Sozzani, 2006*), and possibly Queensland, Australia (*Rudman, 2002*), Thailand (*Panyarachun, 2007*) and Tanzania (*Picton, 2002*).

**Natural history**

Found on soft bottom habitats of silty and sandy substrate where it feeds on the thecate hydroid *Macrorhynchia balei* (Nutting, 1906).

**Etymology**

Puti is the Filipino word for white, reflecting the characteristic opaque white patches on the notum of this species. The white triangular patch on the Philippine flag stands for equality.

**External morphology**

The living animals (Figs. 5A–5D) reach 25 mm in length. The body color is generally translucent white with a solid darker patch of opaque white occupying most of the dorsal surface and extending onto the head region in front of the rhinophores. The pattern of white pigment is irregular and does not extend to the lateral margins of the body, leaving areas of translucence laterally. The opaque patches often contain wrinkles. The digestive gland is a grayish brown, appearing like oxidized milk chocolate. The base of the cerata is translucent white, where the gray digestive gland is readily visible, whereas the medial portion is the most extensive and is an opaque white to light gray band. This band is variable in width and at its apical end has another area of transparency through which the digestive gland is visible. This band is followed by another smaller band of diffuse dull to medium blue, and the most distal part is a creamy yellow at the tip of the cerata. Between these three areas on the cerata, there are translucent areas through which the thin black digestive gland is visible. The rhinophores are thick and dark brown with red color tones that bear a resemblance to kelp leaves. The oral tentacles are the same color as the rhinophores as is the anterior end of the head at the base of the oral tentacles. The rhinophores are smooth, thick, and conical, slightly longer than the narrower oral tentacles. The anterior end of the foot is simply rounded to angular. The cerata project outwards randomly and cover most of the notum.

The cerata are arranged in numerous linear rows. There are 5–6 rows in the precardiac ceratal rows. In one specimen (NMP 041346), the precardiac rows, beginning with the

most anterior row, contain 1, 2, 5, 3, 4, 1 cerata per row. After the interhepatic space, there are six to 13 postcardiac ceratal rows, each of which contains 1–4 cerata. The anus is dorsal and acleioproctic and it is located anterior to the first ceratal row of the postcardiac cerata. The genital opening is ventral to the third and fourth precardiac ceratal rows.

**Buccal mass and radula**

The jaws (Fig. 6A) are dark brown in color and relatively thickly cuticularized with a thickened area on the anterior side of the jaw. The masticatory margin is elongated and contains numerous irregular triangular denticles (Fig. 6B) along its edge. The denticles have fine granular edges. The radula is elongated and tapers considerably with the newest teeth becoming three times as wide as the oldest teeth. In one specimen (NMP 041346) the radula contains a single row of 64 teeth. The central cusp is somewhat shorter than the adjacent lateral denticles and in the oldest teeth (Fig. 6E) is markedly shorter than in the newest teeth (Fig. 6C). There are 4–5 primary lateral cusps on either side of the central cusp. Usually, there are 1–4 secondary denticles between the primary lateral cusps. The posterior end of the tooth is narrow with long spurs extending away from the cutting edge of the tooth (Figs. 6D–6E).

**Reproductive system**

The reproductive system is androdiaulic (Fig. 4B). The ovotestis follicles contain a large female acinus surrounded by a series of smaller male acini. The large, saccate ampulla divides distally into the short oviduct and vas deferens. The vas deferens begins with a thick prostatic portion of and narrows into a short, convoluted ejaculatory duct, entering the penis near the junction of the penial gland with the penial papilla. The penial gland is pyriform whereas the penial papilla is conical with a short, straight, cuticular penial stylet (Fig. 6F). The female glands are well-developed and small albumen and membrane glands are clearly visible, as is the larger mucous gland. A spherical bursa copulatrix is present at the distal end of the reproductive system and connects to the gonopore *via* an elongate duct.

**Remarks**

Our phylogenetic and species delimitation analyses clearly indicate that *T. puti* sp. nov. is distinct from *T. yamasui*. There is a minimum divergence of 18.1% in the COI between *T. puti* sp. nov. and *T. yamasui* and a 12.5% divergence in the 16S rRNA gene. The two species appear to be considered the same species by *Rudman (2002)*, without explanation but are readily distinguishable by the external color pattern and by aspects of their internal anatomy. Externally, the color pattern of *T. puti* sp. nov. is very different from that of *T. yamasui*. In *T. puti* sp. nov., the cerata have opaque white pigment basally, with yellow, black, and blue bands located more apically (Fig. 2), in contrast to the orange cerata with white speckling found in *T. yamasui* (Figs. 5A–5D). Also, *T. yamasui* has bright orange and white pigment on the head that is not present in *T. puti* sp. nov. Additionally, *T. puti* sp. nov. has broad areas of opaque white on the notum that is absent in *T. yamasui*. Internally, the masticatory border of the jaw has denticles with jagged secondary denticles that are not observed in *T. puti* sp. nov. (Fig. 3B). The radular teeth of *T. yamasui* also have

a central cusp that is longer than the adjacent denticles (Fig. 3D), whereas the central cusp is shorter than the adjacent denticles in *T. puti* sp. nov. (Figs. 6C–6E). Furthermore, the teeth of *T. yamasui* lack shorter denticles between the large denticles (Fig. 3D) whereas, *T. puti* sp. nov. has smaller denticles between the larger ones (Figs. 6C–6E). The reproductive system of *T. yamasui* (Fig. 4A) has a relatively short vas deferens that is only slightly convoluted in contrast to the much longer, highly convoluted vas deferens of *T. puti* sp. nov. (Fig. 4B). Similarly, the penial gland of *T. yamasui* has a relatively short duct of the penial gland (*Hamatani, 1993*; present study: Fig. 4A) in contrast to *T. puti* sp. nov. elongate convoluted duct. Ecologically, the two species are found in different habitats. *Tenellia yamasui* is characteristic of living reefs and coral rubble habitats while *T. puti* sp. nov. is found on isolated bits of rock and rubble in soft sediment habitats and feeds on different species of aglaopheniid hydroids.

### *Tenellia bughaw* Kim & Gosliner sp. nov.

LSID:urn:lsid:zoobank.org:act:04C4F439-5D0C-46AD-8245-CDD49BF15C0F

Figures 4C, 5E–5H, 7

*Cuthona yamasui Hamatani, 1993*: *Gosliner, Behrens & Valdés, 2008*: 226, second row left photo, misidentification.
*Cuthona* sp. 13: *Gosliner, Valdés & Behrens, 2015*: 345, right photo.
*Cuthona* sp. 14: *Gosliner, Valdés & Behrens, 2015*: 345, left and right photos.
*Tenellia* sp. F: *Cella et al., 2016*.
*Tenellia* sp. 15: *Gosliner, Valdés & Behrens, 2018*: 287, right photo.
*Tenellia* sp. 16: *Gosliner, Valdés & Behrens, 2018*: 287, left and right photos.

**Holotype**
CASIZ 176737, one specimen, sequenced, St. M24, 2.720101°N. 104.194977°E, Waterfall Bay (S end of the island), near Cahaya Asah Resort, Pulau Tioman, Malaysia, South China Sea, 15m depth, 4 October, 2007, T. Gosliner.

**Paratypes**
CASIZ 176739, one specimen, sequenced, ST. M 09, 3.25569°N, 103.76038°E, Pulau Varella, Malaysia, South China Sea, 15 m depth, 1 October, 2007, T. Gosliner. CASIZ 177552, one specimen, sequenced and dissected, St 23, 13.6864°N, 120.8954°E, Mainit Point, Mabini Batangas, Luzon, Philippines, 21 March, 2008, T. Gosliner. CASIZ 181298, one specimen, St. 14, 3.6859167°N 120.8952167°E, Mainit Bubbles dive site, Mabini, Batangas, Luzon, Philippines, 18 May 2009, Charles Delbeek. CASIZ 200553, one specimen, sequenced and dissected, St. MAB64, 13.65676°N, 20.89682°E, 64, Coconut Point, black Coral Forest, Tingloy, Batangas Province, Luzon, Philippines, 5 May 2014, Alexis Principe.

**Distribution**
Known from eastern Malaysia and the Philippines (present study).

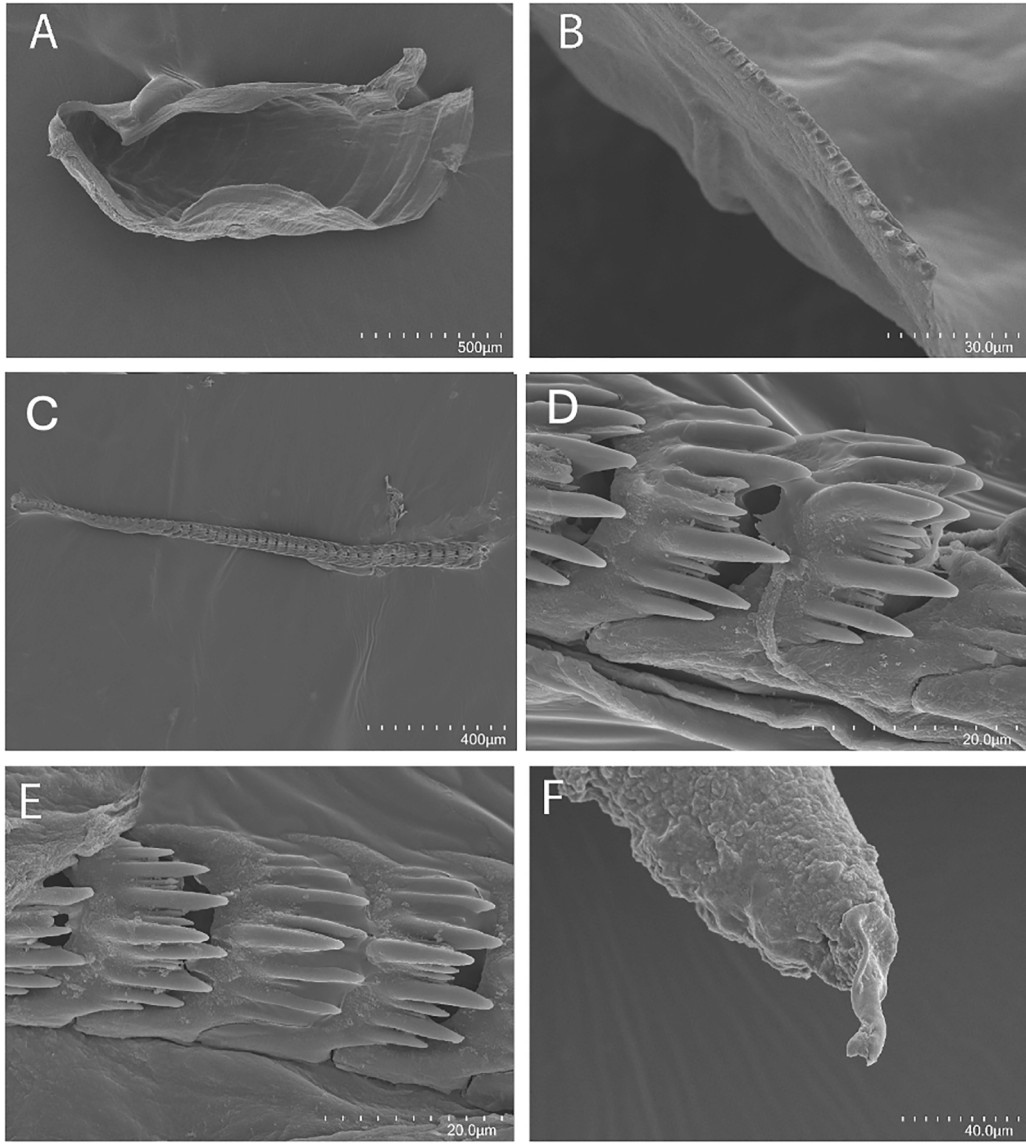

**Figure 7** *Tenellia bughaw* Kim & Gosliner sp. nov., scanning electron micrographs of jaws and radula, CASIZ 186220. (A) Entire jaw. (B) Masticatory border with individual denticles. (C–E) Radular teeth. (F) Penial stylet.

### Natural history
Found on soft bottom habitats of silty substrate where it is found on small bits of rock and rubble where it feeds on the thecate hydroid *Macrorhynchia balei* (Nutting, 1906).

### Etymology
Bughaw is the Filipino word for blue, referring to the bright blue pigment that is dominant in this species.

### External morphology
The living animals (Figs. 5E–5H) reach 25 mm in length. The overall body color is a translucent white with a light blue or purple ombre shade on the dorsal area of the head.

The rhinophores are black or deep plum, but the tip is light blue to white in color. The oral tentacles are very dark plum to black with a small white apex. The cerata are mostly translucent bluish-white, like the body color, but at the tip, there is a grayish-blue shading. Below the tip of each ceras is a medium yellow (with orange tones) band that has a crisp black band above and below it. Also, the cerata are the widest in the medial portion making them appear plump. The rhinophores are slightly shorter than the oral tentacles. They are a little thicker at the base. The tips of the rhinophores are smooth. The anterior end of the foot is simply rounded to angular. The cerata are arranged in numerous linear rows. There are 6–7 rows in the precardiac ceratal rows. In one specimen (CASIZ 200553), the precardiac rows beginning with the most anterior row contain 2, 3, 3, 4, 3, 5, 4 cerata per row. After the interhepatic space, there are seven to 10 postcardiac ceratal rows, each of which contains 1–4 cerata. The anus is dorsal and acleioproctic and it's located anterior to the first ceratal row of the postcardiac cerata. The genital opening is ventral to the third and fourth precardiac ceratal rows.

**Buccal mass and radula**
The concave jaws (Fig. 7A) are brownish and roughly ovate in shape. The masticatory border (Fig. 7B) bears numerous well-spaced triangular denticles. In one specimen (CASIZ 200553) the radula contains a single row of 54 teeth. Each tooth (Figs. 7C–7E) is relatively narrow with a well-elevated central cusp. The central cusp is shorter than the adjacent lateral denticles. The lateral denticles in the middle are longer than the adjacent lateral ones. There are 0–4 secondary denticles located between the primary denticles. They are considerably shorter and narrower than the primary ones. The primary denticles are relatively straight and are acutely pointed to slightly rounded. There are 4–6 primary lateral cusps on either side of the central cusp.

**Reproductive system**
The reproductive system is androdiaulic (Fig. 4C). The overall appearance is lobate and sharply curved. The ovotestis follicles contain a large female acinus surrounded by a series of smaller male acini. The large, saccate ampulla divides distally into the short oviduct and vas deferens. The vas deferens begins with a thick prostatic portion of and narrows into a short, convoluted ejaculatory duct, entering the penis near the junction of the penial gland with the penial papilla. The penial gland is pyriform, whereas the penial papilla is conical. The penial stylet has a tip shape that resembles a floss pick (Fig. 7F), while the base of the stylet has a curved shape. The female glands are well-developed and small albumen and membrane glands are clearly visible, as is the larger mucous gland. A spherical bursa copulatrix is present at the distal end of the reproductive system and connects to the gonopore *via* an elongated duct.

**Remarks**
In our phylogenetic analysis, *Tenellia bughaw* sp. nov. is most closely related to *T. puti* sp. nov.; however, there is a strong genetic divergence between the two species. The minimum genetic divergence between the two species is 14.1% in the COI gene and 9.5% in the 16S rRNA gene. This exceeds the amount of genetic divergence that *Korshunova et al. (2019)*

found for closely related species of *Tenellia* (as *Trinchesia*) where divergence for COI ranged from 2.98–12.99%. Externally, the two are similar in appearance and sometimes difficult to distinguish. The two species are both present sympatrically in the Philippines and have been collected from the same dive site at the same time. Specimens from the Philippines have a more bluish color (Figs. 5F, 5H), whereas Malaysian specimens have a more greenish-white color (Figs. 5E, 5G). Unlike *T. puti* sp. nov., specimens of *T. bughaw* sp. nov. have a uniform frosting of white or blue pigment. Specimens of *T. puti* sp. nov. have patches of white pigment that do not extend to the notal margins. The coloration of the cerata is also distinctive between the two species. In *T. puti* sp. nov., the basal two-thirds of the cerata are covered with opaque white followed by blue subapical band. Above the blue band, near the apex is a succession of diffuse black followed by a band of yellow and another diffuse black area at the very tip. In contrast, in *T. bughaw* sp. nov., the basal two-thirds of the cerata is white, occasionally with a gradual transition to blue. Above the white and blue, there is a sharply defined black band, followed by the yellow-orange band and a second well-defined black band with a white ceratal apex. Internally, the radula of *T. bughaw* sp. nov. (Figs. 7D, 7E) appears to have narrower teeth with more abundant secondary denticles between the primary ones than is found in *T. puti* sp. nov. (Figs. 6C–6E). The reproductive systems of the two species also have several differences (Figs. 4B, 4C). In *T. bughaw* sp. nov., the vas deferens is longer and more highly convoluted than in *T. puti* sp. nov. and the penial gland of *T. bughaw* sp. nov. is proportionately larger than in *T. puti* sp. nov.

***Tenellia nakapila* Kim & Gosliner sp. nov**.

LSID:urn:lsid:zoobank.org:act:C6697544-2DBD-41E3-B00E-9F47C39BB411

Figures 4D, 8A–8D, 9A–9F

*Cuthona* sp. 14: *Gosliner, Behrens & Valdés, 2008*, 364.
*Cuthona* sp. 18: *Gosliner, Valdés & Behrens, 2015*, 346.
*Tenellia* sp. 20: *Gosliner, Valdés & Behrens, 2018*: 288.

**Holotype**
NMP 041347 (formerly CASIZ 202110), one specimen, sequenced, St. CAL03, 13.91422°N 120.60643°E, Lago de Oro house reef, Calatagan, Batangas, Luzon, Philippines, 9 May 2014, T. Gosliner.

**Paratypes**
CASIZ 071139, one specimen, St. 90, Pig (Tab) Island, Madang Papua New Guinea, 10 m depth, January, 1988, T. Gosliner. CASIZ 110386, one specimen, St. 28, Club Ocellaris, Mabini, Batangas Province, Luzon, Philippines, 24 April, 1997, Clay Carlson. CASIZ 208184, one specimen, Airport Beach, Maui, Hawaiian Islands, 8 May 2015. Cory Pittman. CASIZ 208511, one specimen, St. GAL 81, 13.51664°N. 120.95917°E, Schoolhouse Beach, Batangas Channel, Puerto Galera, Mindoro Oriental, Philippines, 13 April, 2015, T. Gosliner. CASIZ 208513, St. GAL91, 13.51688°N, 120.95983°E, Schoolhouse beach,

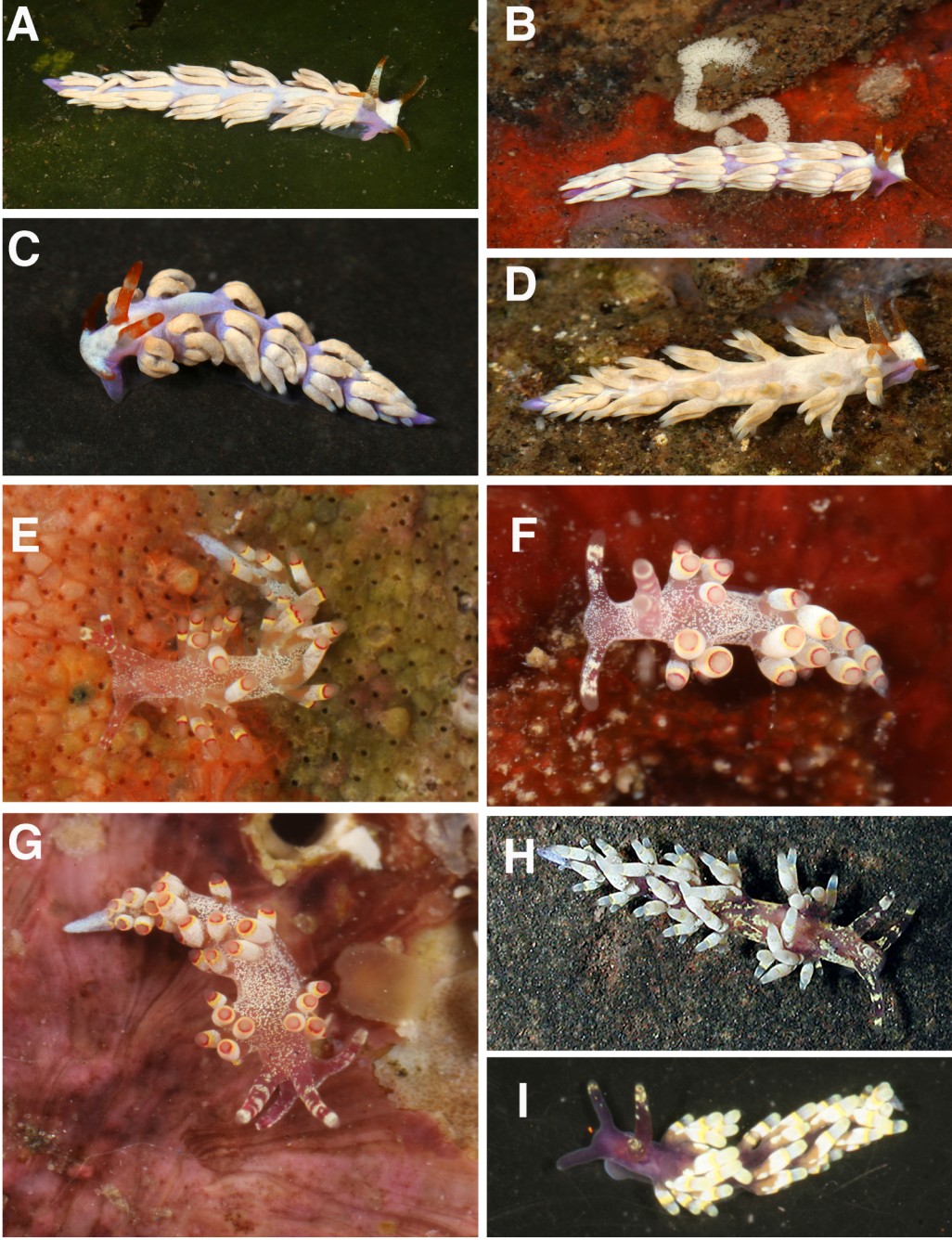

**Figure 8 Living animals, *Tenellia nakapila*, *Abronica payaso*, *A. turon*.** (A) *Tenellia nakapila* Kim & Gosliner, sp. nov., paratype, CASIZ 217310, San Miguel, Dauin, Negros Oriental, Philippines. (B) *Tenellia nakapila* Kim & Gosliner, sp. nov., CASIZ 217303, San Miguel, Dauin, Negros Oriental, Philippines. (C) *Tenellia nakapila* Kim & Gosliner, sp. nov., CASIZ 217311, house reef, Atlantis Resort, Dauin, Negros Oriental, Philippines. (D) *Tenellia nakapila* Kim & Gosliner, sp. nov, CASIZ 217299, San Miguel, Dauin, Negros Oriental, Philippines. (E) *Abronica payaso* Kim & Gosliner, sp. nov. CASIZ 177353, Bethlehem, Tingloy, Batangas, Luzon, Philippines. (F) *Abronica payaso* Kim & Gosliner, sp. nov., CASIZ 208690, Schoolhouse beach, Batangas Channel, Puerto Galera, Mindoro Oriental, Philippines. (G) *Abronica payaso* Kim & Gosliner, sp. nov., CASIZ 208635, Schoolhouse beach, Batangas Channel, Puerto Galera, Mindoro Oriental, Philippines. (H) *Abronica turon* sp. nov., CASIZ 179946, Airport Beach (Kahekili Beach Park), Maui, Hawai'i,. (I) *Abronica turon* Kim & Gosliner sp. nov., CASIZ 191474, Madang lighthouse, Madang Papua New Guinea, all photos by T. Gosliner.

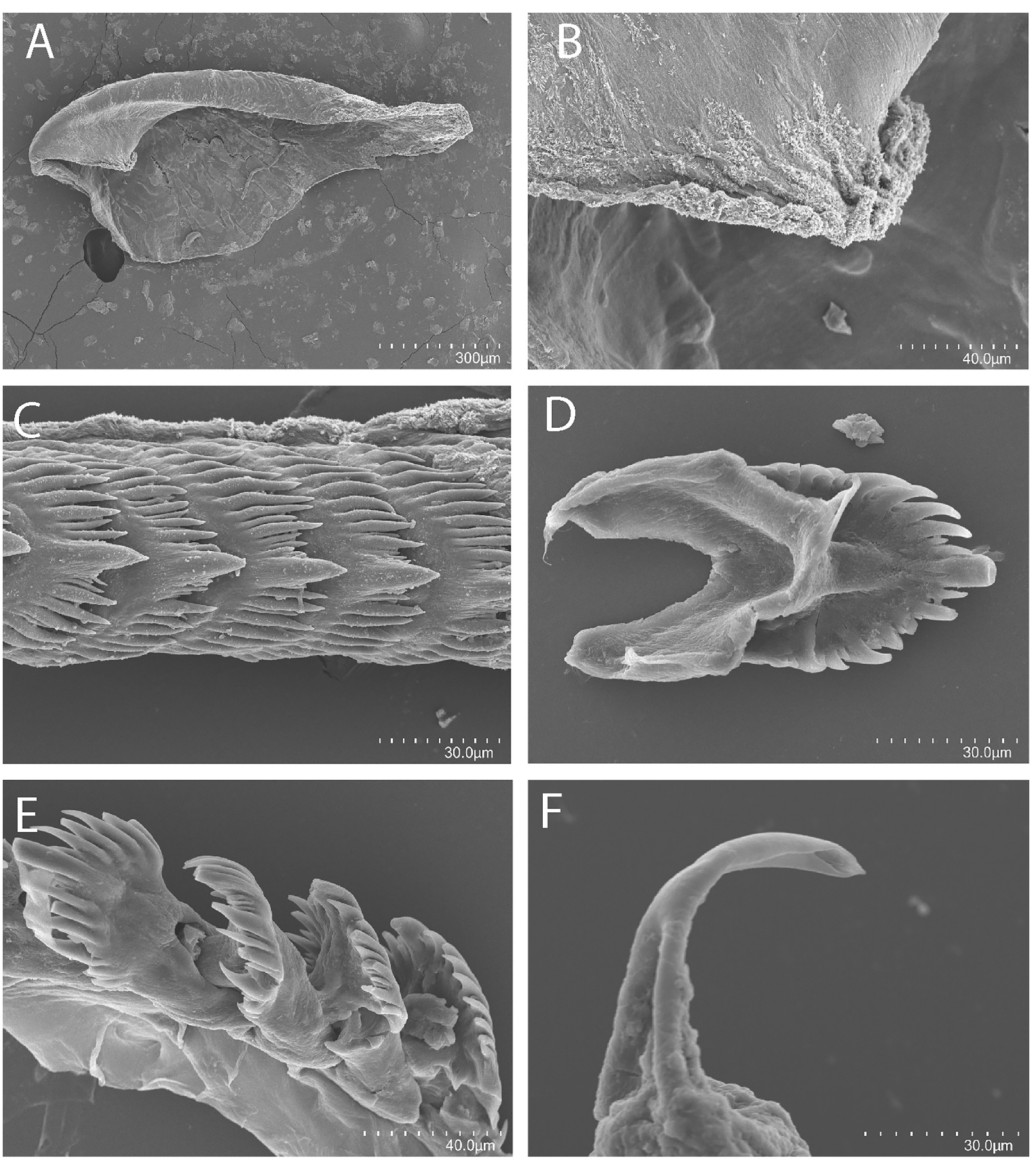

**Figure 9** *Tenellia nakapila* **Kim & Gosliner, sp. nov., scanning electron micrographs of radula, CASIZ 217318.** (A). Entire jaw. (B) Masticatory border with individual denticles. (C–E). Radular teeth. (F) Penial stylet.

Batangas Channel, Puerto Galera, Mindoro Oriental, Philippines, 16 April, 2015, T. Gosliner. CASIZ 208648, one specimen, St. GAL52, 13.51688°N, 120.95983°E, Schoolhouse beach, Batangas Channel, Puerto Galera, Mindoro Oriental, Philippines, 8 April, 2015, T. Gosliner. CASIZ 217318, CASIZ 208579, one specimen, sequenced, St. GAL15, 13.52482°N, 120.97139°E, La Laguna Pt., Puerto Galera, Mindoro Occidental, Philippines, mixed coral and sand, 20 m depth, 27 March, 2015, T. Gosliner. CASIZ 217292, one specimen, ST DAU 02, 9.1864°N, 123.2685°E, Ceres dive site, Dauin, Negros Oriental, Philippines, 15 m depth, 02 April 2016, T. Gosliner. CASIZ 217295, one specimen, St. DAU05, 9.1798°N, 123.259°E, Ginama-An dive site, Dauin, Negros Oriental,

Philippines, 2 April 2016, T, Gosliner. CASIZ 217299, one specimen, St. DAU06, 9.2007°N, 123.2789°E, San Miguel, Dauin, Negros Oriental, Philippines, 2 April, 2016, T. Gosliner. CASIZ 217303, one specimen, sequenced and dissected, St. DAU07, 9.2007°N, 123.2789°E, San Miguel, Dauin, Negros Oriental, Philippines, 3 April, 2016, T. Gosliner. CASIZ 217310, St. DAU13,9.2007°N, 123.2789°E. San Miguel, Dauin, Negros Oriental, Philippines, 9 April, 2016, T. Gosliner. CASIZ 217311, one specimen, St. DAU01, 9.1975°N, 23.2754°E, house reef, Atlantis Resort, Dauin, Negros Oriental, Philippines, 1 April, 2016, T. Gosliner. CASIZ 217313, St. DAU08, 9.1864°N, 123.2685°E, Ceres dive site, Dauin, Negros Oriental, Philippines, 15 m depth, 04 April 2016, M. Burke. CASIZ 217318, five specimens, with eggs, St. DAU16, 9.1968°N, 23.2752°E, VIP resort house reef, Dauin, Negros Oriental, Philippines, 9 April 2016, T, Gosliner & M. Burke.

### Distribution
Known from the Hawaiian Islands Japan, the Philippines and Papua New Guinea (*Gosliner, Behrens & Valdés, 2008*).

### Natural history
Living animals are found on soft sandy to silty substrate where they feed on specimens of the solitary athecate hydroid, *Corymorpha* sp.

### Etymology
The name is derived from the Filipino word nakapila meaning, in a straight line, referring to the very distinctly separated rows of cerata.

### External morphology
The living animals have a narrow, elongated body and reach a length of 7–8 mm. The body color of *Tenellia nakapila* sp. nov. is opaque white on the dorsal side of the anterior end. A lighter purple color in a "V" shape stretches on the dorsal side of the rest of the body. The oral tentacles are a fiery orange-red color with darker yellow speckles. The color perfectly camouflages with the substrate color. The rhinophores are a lighter shade compared to the oral tentacles and have a prominent yellow band near the tip. The anterior end of the foot is simply rounded and somewhat thickened. The cerata base has a dusty light brown color. The tip color is a very bright fiery red. The overall color of the cerata is a light sand color. The cerata color most proximal to the body varies throughout; some are sandy or creamy white. When resting, the animal holds its cerata folded flat against the body. The rhinophores are slightly shorter than the oral tentacles. They are a little thicker at the base. The tips of the rhinophores are sharply pointed to bluntly rounded. The cerata are arranged in numerous linear rows. There are three rows in the precardiac ceratal rows. In one specimen (CASIZ 217303), the precardiac rows beginning with the most anterior row contain 4, 6, 5 cerata per row. After the interhepatic space, there are four postcardiac ceratal rows, each of which contains 1–5 cerata. The anus is dorsal and acleioproctic and it's located anterior to the first ceratal row of the postcardiac cerata. The genital opening is ventral to the third and fourth precardiac ceratal rows.

**Buccal mass and radula**

On either side of the buccal mass are large, dendritic oral glands that extend posteriorly two-thirds of the body length. There is also a pair of short salivary glands. In one specimen (CASIZ 217318), the surface of the jaw has a smooth texture. The jaws (Fig. 9A) are dark brown in color and relatively thickly cuticularized with a thickened area on the anterior side of the jaw. The masticatory margin is elongated and contains numerous irregular denticles (Fig. 9B) along its edge. The radula is elongated and the central cusp is about four times as large as any other denticle. In one specimen (CASIZ 217318), the radula contains a single row of 27 teeth. There are a couple of secondary denticles toward the middle of each tooth (Figs. 9C–9E). There are 5–8 primary lateral cusps on either side of the wider central cusp. Usually, there are 1–4 secondary denticles between the primary lateral cusps. The lateral cusps look like imperfect icicles.

**Reproductive system**

The reproductive system is androdiaulic (Fig. 4D). The overall shape is irregularly ovoid. The ovotestis follicles contain a large female acinus surrounded by a series of smaller male acini. The large, saccate ampulla divides distally into the short oviduct and vas deferens. The vas deferens begins with a thick prostatic portion of and narrows into a short, convoluted ejaculatory duct, entering the penis near the junction of the penial gland with the penial papilla. The penial stylet has a very prominent curved shape, resembling a hook. The base of the stylet is considerably thicker than the tip and looks like a tree trunk (Fig. 9F). The female glands are well-developed and small albumen and membrane glands are clearly visible, as is the larger mucous gland. A spherical bursa copulatrix is present at the distal end of the reproductive system and connects to the gonopore *via* an elongated duct.

**Remarks**

*Tenellia nakapil*a sp. nov. is easily distinguishable by its pale body color and distinctly separated ceratal rows that are held close to the notum. This species is a member of the clade that included coralivorous species formerly placed in the distinct genus *Phestilla*. Our molecular analyses recovered *T. nakapila* sp. nov. as sister to *Tenellia chaetopterana*, which was previously described by *Ekimova, Deart & Schepetov (2019)* from polychaete tubes in Vietnam. This species, *T. chaetopterana*, was recovered as a member of *Phestilla* but with some unusual morphological features. *Fritts-Penniman et al. (2020)* found that *T. chaetopterana* was nested in clade D with other corallivorous species, despite the ecological divergence of this species from its other close relatives. *Tenellia chaetopterana*, like its close corallivorous relatives, has thin, elongated denticles on its radular teeth, but unlike its corallivorous relatives, *T. chaetopterana*, may have cnidosacs that contain nematocysts. In contrast, *T. chaetopterana* sister species, the newly described *Tenellia nakapila* sp. nov., and the undescribed *Tenellia* sp. 58 both feed on hydroids but are also members of the largely corallivorous clade. *Tenellia nakapila* and *T.* sp. 58 are also divergent in that adult have cnidosacs that contain fully functional nematocysts, whereas

only some adult individuals of *T. chaetopterana* store nematocysts and only some juvenile specimens of *T. lugubris* have functional nematocysts (*Putz, König & Wägele, 2010*). *Tenellia nakapila* has radular teeth with lateral cusps that are more typical of other species of *Tenellia* represented in other clades of the phylogenetic tree, whereas *T. chaetopterana* has elongate lateral denitcles more typical of other "coral eating" members of the clade (*Rudman, 1981*). Additionally, *Tenellia nakapila* sp. nov. differs markedly in color from *T. chaetopterana* and *Tenellia* sp. 58 (*Gosliner, Valdés & Behrens, 2018*: 295, middle left image). It also clearly has demarcated rows of cerata that are held close to the body, rather than being held upright as seen in *Tenellia* sp. 58 or *T. chaetopterana*. The three species also differ genetically since there is a strong genetic divergence in the COI/16S rRNA genes of 17.5%/10.3% and 20.3%/9.3% between *T. nakapila* sp. nov. and *Tenallia* sp. 58 and *T. chaetopterana*, respectively.

### *Abronica payaso* Kim and Gosliner, sp. nov.

LSID:urn:lsid:zoobank.org:act:8A25A6A9-9921-4709-B92B-48CAF67E57CE

Figures 4E, 8E–8G, 10A–10F

*Cuthona* sp. 6: *Gosliner, Behrens & Valdés, 2008*: 362.
*Cuthona* sp. 39: *Gosliner, Valdés & Behrens, 2015*: 350.
*Abronica* sp. 3: *Gosliner, Valdés & Behrens, 2018*: 282.

### Holotype
NMP 041348 (formerly CASIZ 177417), one specimen, sequenced, St. 25, 13.673313°N, 120.841566°E, Bethlehem, Tingloy, Batangas, Luzon, Philippines, 19 March 2008, T. Gosliner.

### Paratypes
CASIZ 070430, one specimen, St. 13, −5.1715212°S, 145.842062°E, Barracuda Pt., Madang Papua New Guinea, 13 January 1988, T. Gosliner. CASIZ 070454, one specimen, St. 80, −4.836487°S, 145.783869°E, the Quarry, near Bunn Village, Madang, Papua New Guinea, 11 February 1988, T. Gosliner. CASIZ 071148, one specimen, St. 35, −5.1719232°S, 145.822717°E, Cement Mixer Reef, Madang Papua New Guinea, 19 January 1998, T. Gosliner. CASIZ 071218, one specimen, St. 56, −5.212653°S, 145.815276°E, off of Madang lighthouse, Madang Papua New Guinea, 1 February 1988. T. Gosliner. CASIZ 072769, one specimen, −5.1719232°S, 145.822717°E, Cement Mixer Reef, Madang Papua New Guinea, 20 October, 1986, T. Gosliner. CASIZ 072771, one specimen, St. 60, −5.1719232°S, 145.822717°E, Cement Mixer Reef, Madang Papua New Guinea, 18 October 1986, T. Gosliner. CASIZ 072981, one specimen, St. 33, −5.1715212°S, 145.842062°E, Barracuda Pt., Madang Papua New Guinea, 27 July, 1989, T. Gosliner. CASIZ 072985, 6 specimens, St. 88, −5.1715212°S, 145.842062°E, Barracuda Pt., Madang Papua New Guinea, 27 August 1989, T. Gosliner. CASIZ 072986, four specimens, St. 94, −5.1715212°S, 145.842062°E, Barracuda Pt., Madang Papua New Guinea, 7 m depth, 31 August 1989, T. Gosliner. CASIZ 072991, one specimen, St. 39, −5.164299°S, 145.838340°E, off of the north end of

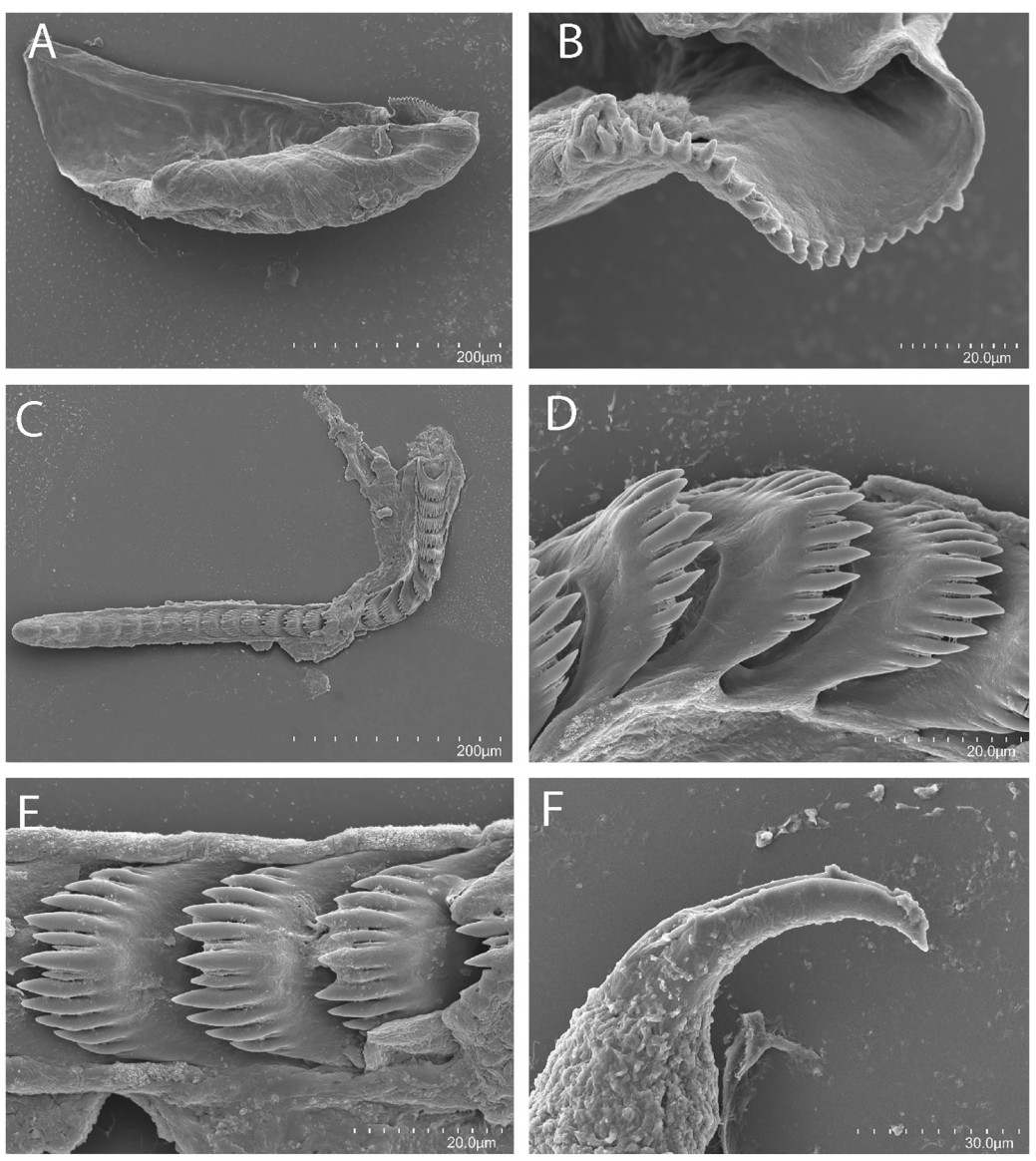

**Figure 10** *Abronica payaso* **Kim & Gosliner, sp. nov., scanning electron micrographs of radula, CASIZ 086449.** (A) Entire jaw. (B) Masticatory border with individual denticles. (C) Entire radula. (D, E) Radular teeth. (F) Penial stylet.

Pig (Tab) Island, Madang, Papua New Guinea, 30 July 1989, T. Gosliner. CASIZ 073009, St. 33, −5.1715212°S, 145.842062°E, Barracuda Pt., Madang Papua New Guinea, 27, July 1989, T. Gosliner. CASIZ 073010, two specimens, St. 64, −5.1715212°S, 145.842062°E, Barracuda Pt., Madang Papua New Guinea, 11 August, 1989, T. Gosliner. CASIZ 073401, two specimens, −5.1719232°S, 145.822717°E, Cement Mixer Reef, Madang Papua New Guinea, 2 m. depth, 22 October 1986. CASIZ 075223, one specimen, St. 29, −5.1715212°S, 145.842062°E, Barracuda Pt., Madang Papua New Guinea, 14 November 1990, T. Gosliner. CASIZ 083850, St. 32, 13.650515°N, 120.841552°E, Devil's Pt., Maricaban Island, Tingloy, Luzon, Philippines, 26 February, 1992, T. Gosliner. CASIZ 086311, one specimen, St. 33, −5.267982°S, 145.830621°E, Planet Rock, Madang Papua New Guinea, 16 June

1992, T. Gosliner. CASIZ 086449, three specimens, one dissected, St. 21, −5.155241°N, 145.830305°E, south side Rasch Passage, Madang, Papua New Guinea, 11 June 1992, T. Gosliner. CASIZ 088083, one specimen, St. 14, 13.720163°N, 120.873558°E, Koala dive site, Mabini, Batangas Luzon, Philippines, 25 March, 1993, T. Gosliner. CASIZ 106592, one specimen, St. 20, 13.690831°N, 120.889058°E, Twin Rocks, Mabini, Luzon, Philippines, 19 April 1996, T. Gosliner. CASIZ 113672, St. 31, −10.203500, 150.937833, Taodovu Reef, off East Cape, SW end of Goschen Strait) Milne Bay Province, Papua New Guinea, 6 June 1998, T. Gosliner. CASIZ 177350, one specimen, sequenced, St. 8, 13.673313°N, 120.841566°E, Bethlehem, Tingloy, Batangas, Luzon, Philippines, 18 March 2008, T. Gosliner. CASIZ 177353, one specimen, sequenced and dissected, St. 8, 13.673313°N, 120.841566°E, Bethlehem, Tingloy, Batangas, Luzon, Philippines, 18 March 2008, T. Gosliner. CASIZ 191475, one specimen, Pig (Tab) Island, Madang, Papua New Guinea, 30 November 2012, V. Knutson. CASIZ 191609, one specimen, −5.1715212°S, 145.842062°E, Barracuda Pt., Madang Papua New Guinea, 8 December 2012, Jessica Goodheart. CASIZ 208447, one specimen, School Beach Batangas Channel, Puerto Galera, Mindoro Oriental, Philippines, 16 April 2015, T. Gosliner. CASIZ 208635, one specimen, St. GAL 52, 13.51688°N, 120.95983°E, Schoolhouse beach, Batangas Channel, Puerto Galera, Mindoro Oriental, Philippines, 8 April, 2015, T. Gosliner. CASIZ 208690, one specimen, ST. GAL 72, 13.51688°N, 120.95983°E, Schoolhouse beach, Batangas Channel, Puerto Galera, Mindoro Oriental, Philippines, 12 April 2015, P. J. Aristorenas. CASIZ 208709, one specimen, St. GAL 132, 13.51688°N, 120.95983°E, Schoolhouse beach, Batangas Channel, Puerto Galera, Mindoro Oriental, Philippines, 29 April 2015, T, Gosliner.

### Distribution
Midway Atoll, Japan, Papua New Guinea and the Philippines (*Gosliner, Behrens & Valdés, 2008*).

### Natural history
Found on the undersides of coral rubble where this species feeds on thecate hydroids.

### Etymology
The name comes from the Filipino word payaso, meaning clown owing to the colorful ornamentation on the body of this species.

### External morphology
The body color is complex. The body is a translucent peach color covered with many small opaque white speckles. There is a denser concentration of white speckles posterior to the head, making it appear like a white clump. Near the head is a darker peach color. The oral tentacles are the same color as the head with dark red and white spots that are in the shape of sprinkles. The rhinophores follow the same pattern as the oral tentacles but have a yellow band near the tip. The anterior end of the foot is simply rounded. There are no speckles on the cerata. The tip of the cerata have a white, yellow, and dark red band in order from most proximal to the body. The cerata located farther away from the head have

a wider opaque white color band. The cerata point in random directions from the body. The body and cerata blend in very well with the environment. The rhinophores are about the same length as the oral tentacles and are smooth in texture. The cerata are arranged in numerous linear rows. There are three precardiac ceratal rows. In one specimen (CASIZ 086449), the precardiac rows beginning with the most anterior row, contain 1, 3, 3 cerata per row. After the interhepatic space, there are five postcardiac ceratal rows, each of which contains 1–3 cerata. The anus is dorsal and acleioproctic and is located anterior to the first ceratal row of the postcardiac cerata. The genital opening is ventral to the third and fourth precardiac ceratal rows.

**Buccal mass and radula**
In one specimen (CASIZ 086449), the jaw is thin and folded when it was air-dried (Fig. 10A). The jaw texture is smooth with undulating grooves over its surface. The jaws are dark brown in color and relatively thickly cuticularized with a thickened area on the anterior side of the jaw. The masticatory margin is elongated and contains numerous irregular denticles (Fig. 10B) along its edge. In one specimen (CASIZ 086449) the radula contains a single row of 28 teeth (Fig. 10C). The individual teeth are broadly arched and elongated (Figs. 10D, 10F). There are 5–6 primary lateral cusps on either side of the equally wide central cusp. The lateral denticles may be longer than the central cusp. There is a pair of secondary denticles flanking the central cusp.

**Reproductive system**
The reproductive system is androdiaulic (Fig. 4F). The texture of the surface of the reproductive system is lobate and irregular in outline. The ovotestis follicles contain a large female acinus surrounded by a series of smaller male acini. The large, saccate ampulla divides distally into the short oviduct and vas deferens. The vas deferens begins with a thick prostatic portion of and narrows into a short, convoluted ejaculatory duct, entering the penis near the junction of the penial gland with the penial papilla. The penis is readily visible. The penial gland is pyriform whereas the penial papilla is conical. The penial stylet (Fig. 10F) is curved like a rainbow shape. The penis is very similar to *Tenellia nakapila* sp. nov., but it is slightly larger in width and length (Fig. 4E). The female glands are well-developed and small albumen and membrane glands are clearly visible, as is the larger mucous gland. A spherical bursa copulatrix is present at the distal end of the reproductive system and connects to the gonopore *via* an elongated duct.

**Remarks**
*Abronica payaso* sp. nov. is clearly distinct from the other described species of *Abronica*: *A. abronia* and *A. purpureoannulata*. Both of these species have purple bands on the oral tentacles and rhinophores, whereas *A. payaso* sp. nov. has red and white spots and an additional yellowish subapical band on the rhinophores. The most distinctive features of *A. payaso* sp. nov. are the peach body color with numerous opaque white speckles and the opaque white, yellow and red bands on the cerata. In our phylogenetic analysis, *A. payaso* sp. nov. is the sister species to both *A. purpureoannulata* and another species described here, *A. turon* sp. nov. *Abronica abronia* is sister to the remaining species of the genus.

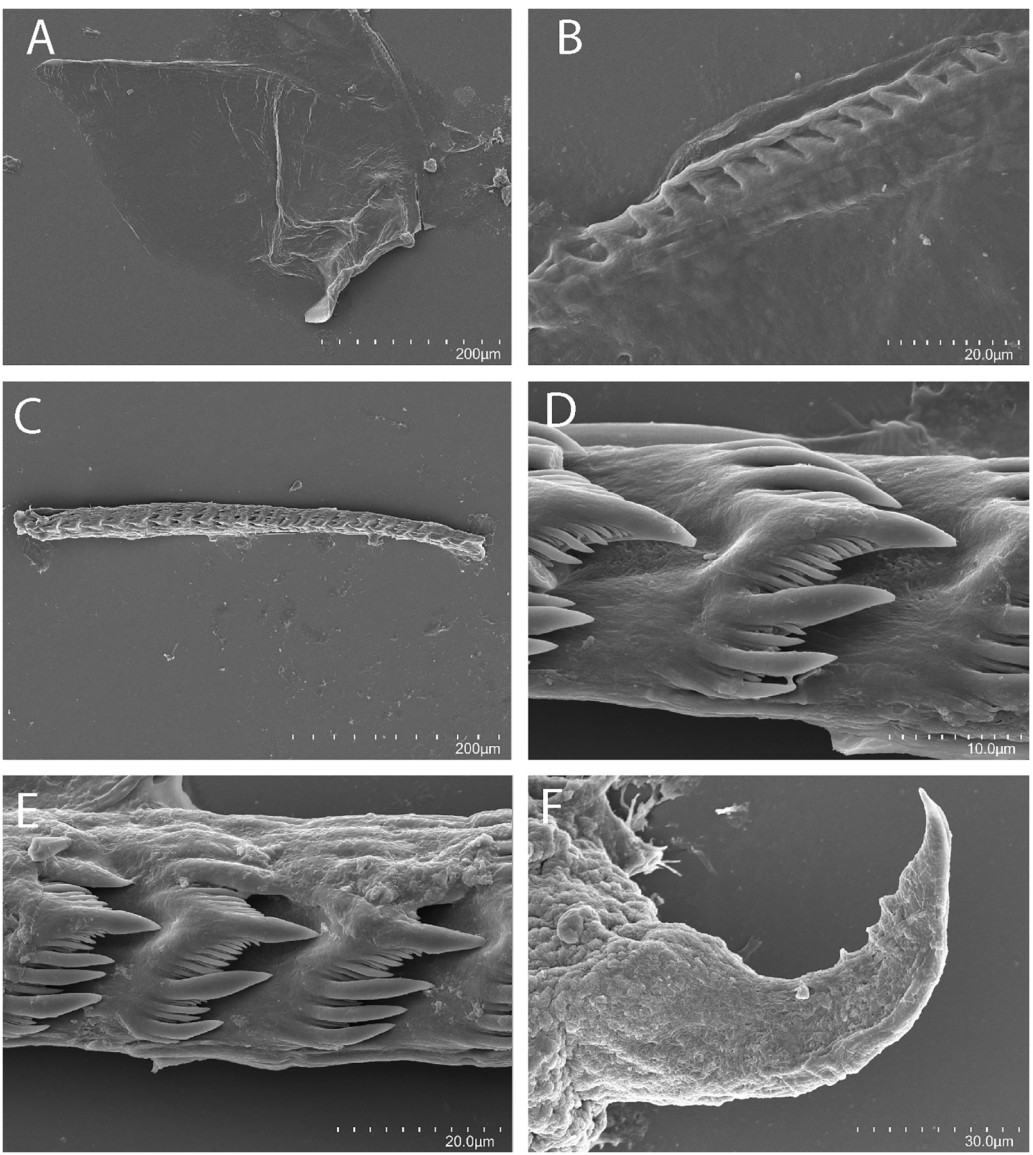

**Figure 11** *Abronica turon* **Kim & Gosliner, sp. nov., scanning electron micrographs of radula, CASIZ 241493.** (A) Entire jaw. (B) Masticatory border with individual denticles. (C) Entire radula. (D, E) Radular teeth. (F) Penial stylet.

Genetically, *A. payaso* sp. nov. is 21.6% different from *A. abronia* in its COI p-distance and 20.5% different from *A. purpureoanulata*.

***Abronica turon* Kim & Gosliner sp. nov**.

LSID:urn:lsid:zoobank.org:act:B3B9E430-20FA-4197-82DD-A0F483D4D240

Figures 4F, 8H, 8I, 11A–11D

*Cuthona* sp. 7: *Gosliner, Behrens & Valdés, 2008*: 362.
*Cuthona* sp. 15: *Gosliner, Valdés & Behrens, 2015*: 346.
*Abronica* sp. 2: *Gosliner, Valdés & Behrens, 2018*: 282.

**Holotype**

CASIZ 179946, one specimen, sequenced, Airport Beach (Kahekili Beach Park), 20.936931°N, −156.694314°W, Maui, Hawai'i, 25 May 2008, C. Pittman.

**Paratypes**

CASIZ 070453, one specimen, S. side Sek Passage, −5.123433°S, 145.823733°E, Madang, Papua New Guinea, 21 February 1988, G. Williamson. CASIZ 071142, one specimen, St. 80, −4.836487°S, 145.783869°E, the Quarry, near Bunn Village, Madang, Papua New Guinea, 11 February 1988, T. Gosliner. 071225, one specimen, St. 41, s. e. side Pig (Tab) Island, Madang, Papua New Guinea, 24 January 1988, R. C. Willan. CASIZ 073001, one specimen, St. 88, −5.1715212°S, 145.842062°E, Barracuda Pt., Madang Papua New Guinea, 31 August 1989, T. Gosliner. CASIZ 073002, St. 94, −5.1715212°S, 145.842062°E, Barracuda Pt., Madang Papua New Guinea, 27 August 1989, M. Ghiselin. CASIZ 073003, St.64, −5.1715212°S, 145.842062°E, Barracuda Pt., Madang Papua New Guinea, 11 August 1989, T, Gosliner. PNG, 073008 St. 14, −5.1715212°S, 145.842062°E, Barracuda Pt., Madang Papua New Guinea, 19 July 1989, T. Gosliner. CASIZ 073514, one specimen, St. 61, −5.1719232°S, 145.822717°E, Cement Mixer Reef, Madang Papua New Guinea, 19 October 1988, T. Gosliner. CASIZ 086397, one specimen, St. 36, −5.155241°N, 145.830305°E, south side Rasch Passage, Madang, Papua New Guinea, 17 June 1992, T. Gosliner. CASIZ 086452, one specimen, St. 12, −5.155241°N, 145.830305°E, south side Rasch Passage, Madang, Papua New Guinea, 8 June 1992, T. Gosliner. CASIZ 086397, one specimen, St. 36, −5.155241°N, 145.830305°E, south side Rasch Passage, Madang, Papua New Guinea, 17 June 1992, T. Gosliner. CASIZ 191474, one specimen, St. PR 135, −5.212653°S, 145.815276°E, off of Madang lighthouse, Madang Papua New Guinea, 29 November, 2012, A. Berberian. CASIZ 241493, one specimen (dissected), St. 28 Daphne's Reef, Madang, Papua New Guinea, 25 July 1989, T. Gosliner. CASIZ 241494, two specimens (one dissected), St. 32, near Madang Lighthouse, Madang, 27 July 1986, T. Gosliner.

**Distribution**

Known from the Hawai'ian Islands and Papua New Guinea

**Etymology**

The name turon comes from the delicious Filipino dessert that is made from fried banana and can be combined with ube (sweet purple yam).

**External morphology**

The overall body color of this species is a dark reddish purple (Figs. 8H, 8I). Their oral tentacles are comparably very long; about one-third of their body length. The main color of the oral tentacles is a very dark purplish blue, almost black color. The tips are bright and greenish-white, like glow-in-the-dark stars. The rhinophores are mostly a dark cherry color with bright greenish-white dots on each one that may be clustered to form a patch. The tips are the same color as the tips on the oral tentacles. On the head are two larger bright greenish-white masses, which are more abundant and widespread in some

Hawai'ian specimens (Fig. 8H). More posteriorly from the head, the body color has an ombre effect that transitions to a dark cherry color. The cerata color medially near the base is a very dark charcoal gray, followed by a thickened area of opaque white which has a honey-yellow band at its apical end. The tips, which abruptly narrow above the opaque white and yellow bands, are a translucent white, fluorescent color. The coloring of this animal makes it appear more fluorescent against their substrate. The direction of the cerata lay somewhat flat against the body or may be slightly elevated. The rhinophores are about half the length of the oral tentacles. The shape is evenly tapered with a somewhat rounded apex. The anterior end of the foot is simply rounded. The cerata are arranged in numerous linear rows. There are two rows in the precardiac ceratal rows. In one specimen the precardiac rows beginning with the most anterior row contain 3, 3 cerata per row. After the interhepatic space, there are 6–7 postcardiac ceratal rows, each of which contains 1–3 cerata. The anus is dorsal and acleioproctic and is located anterior to the first ceratal row of the postcardiac cerata. The genital opening is ventral to the third and fourth precardiac ceratal rows.

**Buccal mass and radula**

In one specimen, the jaw is translucent and flat. The texture has undulating grooves. The jaws (Fig. 11A) are dark brown in color and relatively thickly cuticularized with a thickened area on the anterior side of the jaw. The masticatory margin is elongated and contains numerous regular triangular denticles (Fig. 11B) along its edge. In one specimen (CASIZ 241493), the radula contains a single row of 24 teeth (Fig. 11C). The individual teeth (Figs. 11D, 11E) have a very characteristic shape with three primary denticles on either side of the prominent, acutely pointed central cusp. There are numerous smaller secondary denticles prominently on both sides of the central cusp located in the middle. There are up to nine secondary denticles flanking either side of the primary cusp. Additional secondary denticles are found between the primary denticles. The central cusps are longer than the adjacent denticles.

**Reproductive system**

The reproductive system is androdiaulic (Fig. 4F). The texture of the surface is lobate in the fully mature specimen. The ovotestis follicles contain a large female acinus surrounded by a series of smaller male acini. The large, saccate ampulla divides distally into the short oviduct and vas deferens. The vas deferens begins with a thick prostatic portion of and narrows into a short, convoluted ejaculatory duct, entering the penis near the junction of the penial gland with the penial papilla. The penial gland is pyriform whereas the penial papilla is conical. The penial stylet (Fig. 11F) is very curved and smooth. The base is very thick and tapers outwardly. There are three prominent tubercles on the inner side of the anterior portion of the penial stylet. The female glands are well-developed and small albumen and membrane glands are clearly visible, as is the larger mucous gland. A spherical bursa copulatrix is present at the distal end of the reproductive system and connects to the gonopore *via* a very short duct.

**Remarks**

In our phylogenetic analysis, *Abronica turon* sp. nov. is sister to *A. purpureoanulata* but has a COI divergence of 20.7%. The most characteristic external features of this species are the purple body color with cerata having an abrupt narrowing just apically from the bright yellow band. However, there are also several other internal morphological features that distinguish *A. turon* sp. nov. from the three other known members of the genus. The radular teeth of *A. turon* sp. nov. have numerous secondary denticles whereas *A. abronia*, *A purpureoanulata*, and *A. payaso* sp. nov. lack secondary denticles that are smaller than the primary denticles (*MacFarland, 1966*: pl. 68, figs. 20, 21; *Baba, 1961*: pl. 15, fig. 5; present study: Figs. 10D, 10F). Additionally, *A. turon* sp. nov. is the only species that has a curved penial stylet with three tubercles on its inner edge (fig. 11F). The remaining species have a smooth, curved stylet (*MacFarland, 1966*: pl. 70, fig. 2; *Baba, 1961*: pl. 15, fig. 8; present study: Fig. 10F).

## DISCUSSION

In recent years, there have been major advancements in our knowledge regarding the phylogenetics and systematics of the Fionidae (*sensu lato*) (*Cella et al., 2016*; *Korshunova, Martynov & Picton, 2017*; *Korshunova et al., 2018b*; *Ekimova, Deart & Schepetov, 2019*; *Fritts-Penniman et al., 2020*; *Martynov et al., 2020*; *Korshunova et al., 2020, 2023a*). All of these studies have brought to light multiple additional lineages within the family, but there is strong agreement in the phylogenetic architecture and relationships within Fionidae (*s. l.*). Where there has been considerable disagreement is in how to name the various lineages, and more practically, how much to lump or split this large clade of aeolid nudibranchs. There have been two schools of thought. *Cella et al. (2016)*, *Ekimova, Deart & Schepetov (2019)*, and *Fritts-Penniman et al. (2020)*, have adopted a broad approach based on the integration of molecular and morphological data (lumping), whereas *Korshunova, Martynov & Picton (2017)*, *Korshunova et al. (2018b, 2020, 2023a)* and *Martynov et al. (2020)*, have dramatically proliferated the establishment of new genera and families for the various lineages they have elucidated (splitting), emphasizing what they consider to be important morphological innovations. This fundamental disagreement in taxonomic philosophy has created an unstable and chaotic situation for most systematic biologists and the impacts are far more exacerbated for end users of the classification and taxonomy of nudibranchs including neurobiologists, ecologists, conservation biologists, educators, and community scientists. In making the taxonomic decisions made here we would like to point out what we perceive as several fundamental philosophical and practical problems with the approach followed by the splitters to which we have referenced above.

First of all, there is a practical problem with the splitting approach. One must ask the philosophical question: what is the purpose of taxonomy and classification? Is it to serve the systematist or is it to serve the general public? Even practicing nudibranch taxonomists are now finding it challenging to recognize familial and generic distinctions in this clade and are at a loss to identify a species to a particular genus or family when they encounter a

living specimen in the field or lab. This approach also isolates scientists where there is not equal access to modern tools for the identification of taxa and fails to add to the democratization of science. It also creates additional challenges for the end users of taxonomic information and further reinforces the perception that taxonomists change the names of taxa in an arbitrary and capricious manner. Requiring detailed anatomical and genetic studies to identify a species found in the field further dissociates the general public from these organisms which have a popular following rather than providing an avenue for connecting the general public with the natural world in a manner that they are invited to participate and comprehend. Rather, taxonomists can serve a more useful service in searching for characters that are consistent with their phylogenetic studies that are more simple and accessible to larger audiences, and create classifications that recognize taxa that are recognizable by field characteristics of living organisms and name those lineages.

It can be argued that recognizing new adaptations of a lineage (such as adaptations for burrowing) provides insight into the evolution of that lineage. This however does not require naming a new genus or family every time that a unique innovation occurs, particularly when it can occur multiple times within that lineage (multiple evolutionary events resulting in homoplastic adaptations to the same ecological circumstances). Those novel adaptations can be brought to light by highlighting them on a phylogeny, as was demonstrated by *Martynov et al. (2020*, Fig. 4), but does not require the naming of the additional family Xenocratenidae, for the single species *Xenocratena suecica* Odhner, 1940. Another problem exists with this example. One of the key principles of phylogenetic systematics (*Hennig, 1966*) is that lineages must be based on synapomorphies (shared derived characteristics) rather than uniquely evolved autapomorphies. This is a clear departure from this fundamental principle as is the case of the sister taxon to this species, *Murmania antiqua* Martynov, 2006, another single species elevated to a distinct genus and family based on possession of autapomorphic features. In this example, the sister taxon to Murmaniidae and Xenocratenidae is Cuthonellidae, a family consisting of a single genus with 18 species (*WoRMS, 2023*). Another principle of modern phylogenetic systematics is that taxonomic ranks should have equivalency between sister taxa: both sister taxa should be at the same rank since they represent lineages that are divergent from each other at the same moment in evolutionary history. This principle has clearly not been followed by the practitioners of the splitter school. It appears that the assignment of either generic or familial status has been applied arbitrarily, depending on a subjective evaluation of the degree of morphological divergence or ontogenetic change in the various lineages rather than any application of any consistently objective criteria.

*Korshunova, Martynov & Picton (2017)* were quite critical of the lumping approach first suggested by *Cella et al. (2016)*. They argued that the proposed classification does not provide a solution because it does not include synapomorphies and masks a diverse phylogenetic pattern. This argument is unclear since the phylogeny presented clearly shows a phylogenetic pattern with a distinct pattern of diverse lineages. The other point that morphological synapomorphies for Fionidae (*s.l.*) were not presented is contradicted by their next paragraph where the authors attempt to refute the synapomorphies for Fionidae listed by *Cella et al. (2016)*. Four synapomorphies were presented for the family:

acleioproctic anus, presence of a penial gland, rounded head, and reproductive follicles with peripheral male acini. *Korshunova, Martynov & Picton (2017)* contend that the anus is cleioproctic rather than acleioproctic in several fionid taxa: *Cuthona*, *Murmaina* and the type species of *Cuthonella*, *C. abyssicola* Bergh, 1884. While they criticize the lack of use of ontogenetic data by other workers, this statement largely ignores the work by *Williams & Gosliner (1979)* who clearly demonstrated that the "cleioproctic" condition found in *Cuthona nana* (Alder & Hancock, 1842) is secondary. The original acleioproctic condition of juvenile specimens becomes masked when posterior ceratal rows extend anteriorly in more mature individuals to surround the anus. This resulting anal configuration is clearly distinct from the true cleioproctic condition found in other aeolids (*e.g.*, Aeolidiidae, Myrhinnidae, Facelinidae). In *Cuthonella*, *Korshunova et al. (2020)* presented a phylogeny that indicates that *C. abyssicola* is a more derived member of the clade, indicating that its "cleioproctic" anus is a secondarily derived condition within that lineage. Nothing is known about the ontogenetic and phylogenetic derivation of the anus in *Murmania*. *Korshunova, Martynov & Picton (2017)* also contended that a penial gland cannot be considered a synapomorphy for Fionidae since the penial gland is absent in *Tergiposacca* and *Fiona*. This is tantamount to suggesting that tetrapods cannot be considered to have a synapomorphy of having four limbs, since legs have been lost in caecilians, snakes, and legless lizards, a proposition that few systematists and evolutionary biologists would accept. The authors simply ignore the other two synapomorphies listed by *Cella et al. (2016)* for Fionidae.

*Korshunova, Martynov & Picton (2017*, fig. 7) divided Trinchesiidae into several genera based on the phylogeny depicted and the description of morphological features. However, this treatment is very typological (focusing largely on the type species of each genus) and does not include any explicit analysis of intraspecific or interspecific variation. For example, they characterize species they include in *Phestilla* as very distinctive based on morphology and dietary specialization. They state that species of *Phestilla* lack cnidosacs, but more correctly, the cnidosacs are present and lack functional nematocysts, as in the case of the recently described *T. chaetopterana Ekimova, Deart & Schepetov (2019)*, which is variable in its presence or absence of cnidosacs. Two other species in this clade, *Tenellia* sp. 58 and *Tenellia nakapila* Kim and Gosliner, sp. nov. have well-developed cnidosacs that contain fully functional nematocysts (present study). Both of these species have a typical fionid radula without elongated lateral denticles and feed on hydroids. *Tenellia chaetopterana* is found in chaetopterid polychaete tubes and clearly does not feed on corals, despite being found within clade D of these coral specialists (*Fritts-Penniman et al., 2020*, fig 2). Thus, variations in characteristics that are purported to represent unique attributes of *Phestilla* are not shared by all members of the lineage. If one employs the attributes suggested by *Korshunova, Martynov & Picton (2017)*, *Phestilla* represents a paraphyletic lineage. Similarly, *Korshunova, Martynov & Picton (2017)* state that members of the genus *Trinchesia* "uniquely possess tentacular foot corners". Closer examination reveals that only the type species, *T. caerulea* (Montagu, 1804), has tentacular foot corners, while other species have angular or rounded foot corners (present study). The presence of tentacular foot corners as an apomorphy was further negated by *Korshunova et al. (2019)*,

where different members of a species complex of "Trinchesia" have either tentacular, angular or rounded foot corners. Similarly, the distinctness of *Zelentia* and *Diaphoreolis* is not clearly articulated and no detailed comparison of included species is made. Furthermore, *Diaphoreolis* apparently has no unique characteristics that differentiate it from its sister taxon *Trinchesia*.

*Korshunova et al. (2023a)* in an attempt to resolve the lumper's and splitter's dilemma, suggested a series of rules to be employed when making systematic decisions. First, one should use both morphological and molecular data in informing taxonomic decisions. Nobody would disagree with this point and in fact, better, more informed taxonomic decisions should be made by including all available lines of evidence. This is the very nature of science. The second suggested "rule" is that aberrant taxa nested within larger clades should not be included in the same taxon as the rest of the lineage so as to highlight the unusual morphology of that taxon. As noted above, this has several implications that are problematic. First naming that taxon must not render the other taxa within that lineage paraphyletic. Monophyly must be preserved. Secondly, that aberration must not represent an autapomorphy but must be a true synapomorphy, determined by a rigorous examination through outgroup comparison. Determining what is aberrant or just unusual is very subjective and does not seem to be universally applied. For example, the possession of a "cleioproctic" anus in *Cuthonella abyssicola* as noted by *Korshunova et al. (2020)* while other members of the genus have a typical acleioproctic anus was not regarded as a "noteworthy" aberration, whereas depressed radular cusp of many "*Catriona*" is "noteworthy" What is different enough? Again, a very subjective interpretation must be invoked. "Rule" three says that monotypic genera should not be avoided. However, this is negated by the fact monotypic genera, by definition, contain only a single species and the noteworthy aberration is therefore an autapomorphy, a clear violation of phylogenetic systematic principles.

The fourth "rule" states when molecular evidence suggests the heterogeneous nature of a taxon that it should be separated into several distinct taxa of equal rank. While the meaning of this is ambiguous, what we think the authors are suggesting, based on the example of *Korshunova, Martynov & Picton (2017)* is that *Tenellia* as suggested by *Cella et al. (2016)* should be divided into several genera and families because *Tenellia* is morphologically heterogeneous. However, it has been pointed out that the division proposed by *Korshunova, Martynov & Picton (2017)*, has dubious synapomorphies, does not reflect variation within the lineages they named, and creates a series of taxa where sister taxa are not at the same rank. Again, the decisions of when to create a genus or family are made entirely upon the subjective decision of how different they are morphologically and what is an important difference *vs.* one that is less important. All of this points to subjective arbitrariness. The fifth "rule" suggests that the presence of intermediate conditions should not be a basis for lumping taxa as each taxon has its own ontogenetically-fueled evolutionary history. Natural selection fuels evolution, not ontogeny. The presence of intermediate conditions is relevant if there is a variation that contradicts the suggested synapomorphies that supposedly characterize that taxon. This is a specious argument that is cloaked in evolutionary and ontogenetic rhetoric that has little relevance to the more

important issue that synapomorphies should be real and not just restricted to the typological characterization of a taxon. "Rule" 6 suggests morphologically fine-scale assessment should be done for every differentiated taxon. Again, this is pretty obvious and good species descriptions do this and carefully compare taxa within a larger taxon. However, it should be noted that this approach was not followed by *Korshunova, Martynov & Picton (2017)*, where only the type species of the taxa recognized were characterized and there was no critical exploration of fine-scale morphological variation. Lastly, "rule" 7 states that large-volume genera should be avoided since they obscure both morphological and molecular variation. This seems arbitrary and authoritarian. *Korshunova et al. (2020)* described seven new species of *Cuthonella*, bringing the total known species to 18. When does a small genus become a medium or large one and at what point should it be divided? "Rule" 7 provides no guidance as to how large is large.

*Cella et al. (2016)*, *Epstein et al. (2018)*, and *Donohoo et al. (2023)* all dealt with large genera and the discussion of molecular and morphological variation and unique adaptations constrained by the fact that many species are all included in a single genus. These articles in fact discuss variation and highlight unique innovations. For example, *Epstein et al. (2018)* noted the independent evolution of similar color patterns in *Hypselodoris* and *Donohoo et al. (2023)* highlighted the existence of a clade of *Halgerda* that has differentiated in deep water habitats. They preferred to retain large genera since they are easily characterized and recognized by the general public and end users of taxonomy. There is a strong precedence for doing so. The avian genus *Turdus* includes 86 species, but there is no suggestion to divide it into dozens of genera and multiple families. Other genera in the family are generally smaller in their number of species. Ornithologists appear to be completely comfortable with a single large genus and several smaller genera in a single family. Other large genera include angiosperms in the genera *Astragalus* (more than 3,270 species), *Solanum* (more than 1,500 species arranged in a series of subgenera and sections), *Senecio* (approximately 1,250 species) to name a few other examples. In fact, 57 genera of plants are considered to contain 500 or more species (*Frodin, 2004*). The "rules" presented by *Korshunova et al. (2023a)* create some contradictions with other systematic principles already widely adopted by phylogenetic systematics and also conflate recognizing unique adaptations and morphological attributes with explicitly requiring that names reflect every interesting novelty that is a product of evolution. There is clear agreement that we must include as many lines of evidence as possible in making taxonomic decisions and that highlighting unique adaptations is a key part of systematic biology. But we serve a larger world where taxonomists must create systems of classification that are accessible and understandable by a general population that uses this information, often for very practical purposes.

If we consider our present state of knowledge of the Fionidae, *Cella et al. (2016)* noted that there were at least 70 undescribed species of Fionidae from the Indo-Pacific. Since then, *Gosliner, Valdés & Behrens (2018)* modified that number to include more than 115 undescribed species of Fionidae from the Indo-Pacific. *Fritts-Penniman et al. (2020)* indicated that analysis of a clade of coral-eating species of *Tenellia* recognized 22 species in this lineage rather than five described taxa traditionally recognized taxa. *Korshunova et al.*

*(2020)* noted seven new species of *Cuthonella*, mostly from the boreal northern Pacific. *Korshunova et al. (2023a)* added new species of *Cartriona* and *Tenellia*. In the cases of *Cuthonella*, *Catriona*, and *Tenellia*, new species have been identified within well-established lineages. It should be noted that *Tenellia* (*sensu stricto* of Korshunova et al. 2017) was separated as a distinct genus, since *Tenellia adspersa* has reduced oral tentacles and retains some paedomorphic features such as a rounded head with no distinct oral tentacles. There is little precedent for this as exemplified in the case of the model organism and paedomorphic salamander *Ambystoma mexicana* (axolotl) which is considered to be a close relative of the fully developed species, in the *A. tigrinum* complex. Members of the genus *Ambystoma* are considered in the same genus to reflect their common ancestry, not to highlight the unusual paedomorphic condition of one aberrant taxon (*Shaffer, 1993*). The unusual nature of developmental characters is highlighted through their phylogeny rather than highlighting them through being classified in different higher taxa.

Many other species included in the genus *Tenellia* (*sensu stricto*) appear to represent previously undocumented lineages (*Korshunova, Martynov & Picton, 2017*; present study). Ultimately, naming these lineages may prove practical, once the phyletic diversity, species richness, intraspecific, and interspecific variation of morphological features have been more thoroughly studied. While some individuals appear to be comfortable with the taxonomic inflation of naming new families and/or genera for every new lineage discovered, we prefer to limit higher taxa to those that can be easily recognized by taxonomists and other practitioners of our classification systems, including the general public. We have articulated many of the issues with the extreme splitting approach and placed the new taxa identified and described here within the genera *Abronica* and *Tenellia*. We affirm that we consider the classification system adopted by *Cella et al. (2016)* to better reflect the diversity within Fionidae and as a system of classification that reflects our understanding of newly understood phylogenetic relationships in a way that is the least disruptive to traditional taxonomy. Again, as noted above, this situation is not unique to the study of the fionid clade of nudibranchs. We fully support the separation of taxa when new phylogenetic relationships are revealed as was adopted in the separation of species of *Samla* from the Flabellinidae (*Korshunova et al., 2017*). However, we do not support what we regard as an unnecessary proliferation of higher taxa, which increasingly confuses the end users of taxonomy and increasingly isolates systematic biology from the individuals who use our system of classification. To do otherwise is a disservice to a larger community that will ultimately make taxonomy less relevant and ultimately likely accelerate its extinction.

## CONCLUSION

This article describes five new species of Fionidae (*s.l.*) from the Indo-Pacific tropics. These species are members of three distinct clades of fionids and three of these species appear to be inhabitants of soft-bottom communities adjacent to well-developed coral reefs. All three of these clades include representatives that are present in both temperate and tropical regions. Species of *Abronica*, thus far, are restricted to the temperate Pacific and the

tropical Indo-Pacific. The clade that includes *Tenellia yamasui, T. bughaw* sp. nov. and *T. puti* sp. nov. includes tropical Indo-Pacific and temperate Atlantic and Pacific representatives. The clade that includes *T. nakapila* sp. nov. is thus far restricted to the Indo-Pacific and the majority of species are specialist predators of scleractinian corals. The classification of the Fionidae has recently been very unstable and controversial. We present evidence that supports a more conservative separation of genera, especially given the number of apparently undescribed taxa that are known to exist, as their phylogeny, morphology, and biology remain largely unknown. In our view, a taxonomic subdivision of this large and diverse clade must await a more comprehensive understanding of the evolutionary biology of its members.

## ACKNOWLEDGEMENTS

The authors would like to thank the California Academy of Sciences for the opportunity to work in their advanced facility. We thank Cory Pittman, Jessica Goodheart, and Vanessa Knutson for the collection of specimens. We would also like to thank Michelle Nishijima for her assistance in the specimen naming process.

This collaborative research involved the following partners in the Philippines: former Secretary of Agriculture Proceso J. Alcala; former Philippine Consul General Marciano Paynor and the Consular staff in San Francisco; former Bureau of Fisheries and Aquatic Resources (BFAR) Director Attorney Asis G. Perez; BFAR colleagues, especially Attorney Analiza Vitug, Ludivina Labe; National Fisheries and Research Development Institute (NFRDI) colleagues, especially Director Drusila Bayate and November Romena; U.S. Embassy staff, especially Heath Bailey, Richard Bakewell and Maria Theresa N. Villa; staff of the Department of Foreign Affairs; University of the Philippines (UP) administrators and colleagues including UP President Alfredo Pascual, Vice President Giselle Concepción, Dr Annette Meñez; the staff of the National Museum of the Philippines, especially Dr Jeremy Barns, Anna Labrador and Marivene Manuel Santos. We also thank Jessie de los Reyes, Marites Pastorfide, Sol Solleza, Boy Venus, Joy Napeñas, Peri Paleracio, Alexis Principe, the staff of Atlantis Dive Resort Puerto Galera (especially Gordon Strahan, Andy Pope, Marco Inocencio, Stephen Lamont, P.J Aristorenas), the staff of Lago de Oro Beach Club and Protacio Guest House, May Pagsinohin, Susan Po-Rufino, Ipat Luna, Enrique Nuñez, Jen Edrial, Anne Hazel Javier, Jay-o Castillo, Arvel Malubag and Mary Lou Salcedo. Lastly, our sincere thanks are extended to our fellow Academy and Filipino teammates on the expeditions. This is part of the joint Department of Agriculture-NFRDI-California Academy of Sciences Memorandum of Agreement for the ongoing implementation of the National Science Foundation-funded biodiversity expedition in the Verde Island Passage.

### Funding

This research was supported by a grant from the National Science Foundation: DEB 12576304 grant to Terrence Gosliner, Richard Mooi, Luis Rocha, and Gary Williams and

REU 1358680 to Richard Mooi. Field expeditions to Papua New Guinea were supported by funding from the Total Foundation, Prince Albert II of Monaco Foundation, Foundation EDF, Stavros Niarchos Foundation, and Entrepose Contracting, and in-kind support from the Divine Word University (DWU). The funders had no role in study design, data collection and analysis, decision to publish, or preparation of the manuscript.

### Grant Disclosures
The following grant information was disclosed by the authors:
National Science Foundation: DEB 12576304 and REU 1358680.
Total Foundation.
Prince Albert II of Monaco Foundation.
Foundation EDF.
Stavros Niarchos Foundation.
Entrepose Contracting.
Divine Word University (DWU).

### Competing Interests
The authors declare that they have no competing interests.

### Author Contributions

- Ashley Y. Kim conceived and designed the experiments, performed the experiments, analyzed the data, prepared figures and/or tables, authored or reviewed drafts of the article, and approved the final draft.
- Samantha A. Donohoo analyzed the data, prepared figures and/or tables, authored or reviewed drafts of the article, and approved the final draft.
- Terrence M. Gosliner conceived and designed the experiments, prepared figures and/or tables, authored or reviewed drafts of the article, and approved the final draft.

### Field Study Permissions
The following information was supplied relating to field study approvals (*i.e.*, approving body and any reference numbers):

All the specimens from the Philippines were collected under our Gratuitous Permits (GP-0077-14, GP-0085-15) from the shallow waters of the municipalities of Mabini, Tingloy, Calatagan and Puerto Galera.

### DNA Deposition
The following information was supplied regarding the deposition of DNA sequences:

The sequences are available at GenBank: PP759731–PP759736, PP751617 (16S rRNA); PP747810–PP747814, PP751619 (COI); and PP750950–PP750955, PP768732 (H3).

### Data Availability
The sequences are available at GenBank: PP747810, PP747812-PP747814, PP750950, PP750952-PP750955, PP751617, PP751619, PP759731, PP759733-PP759736.

## New Species Registration

The following information was supplied regarding the registration of a newly described species:

Publication LSID: urn:lsid:zoobank.org:pub:1757F835-C668-4300-961A-FC1E8B409CE7

*Tenellia puti* LSID: urn:lsid:zoobank.org:act:65A96380-78E1-48DC-B344-6D0FC8549A19

*Tenellia bughaw* LSID: urn:lsid:zoobank.org:act:04C4F439-5D0C-46AD-8245-CDD49BF15C0F

*Abronica payaso* LSID: urn:lsid:zoobank.org:act:8A25A6A9-9921-4709-B92B-48CAF67E57CE

*Abronica turon* LSID: urn:lsid:zoobank.org:act:B3B9E430-20FA-4197-82DD-A0F483D4D240

*Tenellia nakapila* LSID: urn:lsid:zoobank.org:act:C6697544-2DBD-41E3-B00E-9F47C39BB411.

## Supplemental Information

Supplemental information for this article can be found online at http://dx.doi.org/10.7717/peerj.18517#supplemental-information.

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
