# Peer review of "Stirring up the muck: the systematics of soft-sediment Fionidae (Nudibranchia: Aeolidina) from the tropical Indo-Pacific"

_PeerJ, doi:10.7717/peerj.18517_

## Round 0.1 · original submission · Major Revisions

We are pleased to inform you that your manuscript has passed the peer review stage and is ready for revision. The reviewers provided detailed comments, and I ask that you consider these carefully when revising the manuscript as well as respond to their suggestions in the cover letter when you re-submit. This will help avoid further rounds of explanations and revisions, and allow quickly move to the decision.

Reviewers found a number of inaccuracies in species names and misidentifications in the Table 1.

The database of phylogenetic analyses needs to be significantly expanded using data from recently published work.

Figure 1 needs to be improved as it is very hard to read. I recommend you to consider the of the Reviewers to reorganize the sections Discussion.

Additionally, Reviewer 2 reacted very emotionally to the idea of different access to modern taxa identification tools for scientists in developing countries compared to scientists in developed countries. Since this phrase may have different meanings, I ask you to exclude it from the Discussion.

Reviewer 1 ·

Basic reporting

The manuscript is well-written, English is clear and professional. The manuscript is well organized and includes relevant background information on Indo-West Pacific diversity of Fionidae s.l. and the discussion of current taxonomy of these nudibranchs. All cited literature is relevant to studied topic, however I feel important to include several additional studies of the Indo-Pacific Tenellia, which were recently published and include novel molecular data and discussion (see specific comments below). Most figures are excellent, however I have some suggestions for improvement of the Figure 1, as tip labels are hard to read (see below).

l. 53-56, 58-60: there are several other relevant papers, describing new taxa of the genus 'Phestilla' from the Indo-Pacific. I suggest including them to the Introduction and the Discussion, as they describing taxa, which are possibly closely related to T. nakapila:
Mehrotra, R., Arnold, S., Wang, A., Chavanich, S., Hoeksema, B. W., & Caballer, M. (2020). A new species of coral-feeding nudibranch (Mollusca: Gastropoda) from the Gulf of Thailand. Marine Biodiversity, 50(3), 36.
Hu, J., Zhang, Y., Yiu, S. K. F., Xie, J. Y., & Qiu, J. W. (2020). A new species of predatory nudibranch (Gastropoda: Trinchesiidae) of the scleractinian coral Goniopora. Zoological Studies, 59.
Wang, A., Conti-Jerpe, I. E., Richards, J. L., & Baker, D. M. (2020). Phestilla subodiosus sp. nov.(Nudibranchia, Trinchesiidae), a corallivorous pest species in the aquarium trade. ZooKeys, 909, 1.
Mehrotra, R., Caballer, M., Kaullysing, D., Jualaong, S., & Hoeksema, B. W. (2024). Parasites or predators? Gastropod ectoparasites and their scleractinian host corals at Koh Tao, Gulf of Thailand, with the description of a new species. Symbiosis, 92(2), 209-230.

Fig 1: The figure is very hard to read, because tip labels are too small. I think that many species-level clades (or even genera you not discussing, like Cuthonella, etc) except your target taxa may be hided ('collapse' option in FigTree) to show only one specimen per each clade, this will reduce the tree length and enable authors to enlarge the font of labels. You can include the full version of the tree in Supplementary material thus any reader will be able to see phylogenetic relationships in details.

Experimental design

The manuscript fully conforms Aims and Scope of the journal and contains novel and sufficient information. The authors use an integrative approach to evaluate the systematics of Abronica and Tenellia. They examine two mitochondrial and one nuclear genes to place studied species into the phylogenetic framework. They support their molecular phylogenetic results by various species delimitation methods, their results are also congruent with morphological data. In result, they describe five new species and their identity is fully justified. Overall I do not have any concerns regarding methodology, as all methods are described in full and all relevant information is provided either in the main text or in the supplementary material. At the same time, I find important to significantly expand the dataset of the phylogenetic analysis using data from recently published papers. I suggest a few mandatory research below, but there are clearly more. Since one of the most important parts of the manuscript is the discussion of Tenellia clade and 'splitting' versus 'lumping' taxonomical scheme of the family Fionidae, a comprehensive dataset with inclusion of all available Tenellia sensu lato species has a crucial importance. Please consider inclusion of all 'Zelentia' spp., 'Diaphoreolis' spp., 'Catriona' spp., Tenellia s.str., 'Phestilla' spp., 'Trinchesia' spp., datasets and sequences were analysed in the following papers:

Korshunova, T., Lundin, K., Malmberg, K., & Martynov, A. (2023). Narrowly defined taxa on a global scale: The phylogeny and taxonomy of the genera Catriona and Tenellia (Nudibranchia, Trinchesiidae) favours fine‐scale taxonomic differentiation and dissolution of the “lumpers & splitters” dilemma. Evolutionary Applications, 16(2), 428-460.

Korshunova, T., Fletcher, K., Bakken, T., & Martynov, A. (2023). The first consolidation of morphological, molecular, and phylogeographic data for the finely differentiated genus Diaphoreolis (Nudibranchia: Trinchesiidae). Canadian Journal of Zoology, 101(8), 635-657.

Korshunova, T., & Martynov, A. (2022). Increased information on biodiversity from the neglected part of the North Pacific contributes to the understanding of phylogeny and taxonomy of nudibranch molluscs. Canadian Journal of Zoology, 100(7), 436-451.

Mehrotra, R., Caballer, M., Kaullysing, D., Jualaong, S., & Hoeksema, B. W. (2024). Parasites or predators? Gastropod ectoparasites and their scleractinian host corals at Koh Tao, Gulf of Thailand, with the description of a new species. Symbiosis, 92(2), 209-230.

Korshunova, T., Picton, B., Furfaro, G., Mariottini, P., Pontes, M., Prkić, J., ... & Martynov, A. (2019). Multilevel fine-scale diversity challenges the ‘cryptic species’ concept. Scientific Reports, 9(1), 6732.

... and others ...

Also, I identified several misidentifications in your Table 1. All these inconsistencies must be checked carefully and corrected according current system.

1) 'Eubranchus alexii' should be ''Eubranchus alexei'
Eubranchus exiguus (Alder and Hancock, 1848)
2) Eubranchus exiguus WS3456 and GnM9092 are in fact Eubranchus scintillans, please see Grishina, D. Y., Schepetov, D. M., & Ekimova, I. A. (2022). Hidden beauty of the north: a description of Eubranchus scintillans sp. n.(Gastropoda: Nudibranchia) from the Barents Sea and North-East Atlantic. Invertebrate Zoology, 19(4), 351-368.
3) Why do you identify Eubranchus sp. 3 as recently described Eubranchus putnami Fernandez-Simon & Moles,
2023? The COI of E. putnami is available in GenBank (OQ206999) and it is clearly different from sequences of Eubranchus sp. 3. This is very confusing.
4) please consider replacing most of your 'Cuthonella soboli' by 'Cuthonella osyoro', please see Korshunova, T. A., Sanamyan, N. P., Sanamyan, K. E., Bakken, T., Lundin, K., Fletcher, K., & Martynov, A. V. (2020). Biodiversity hotspot in cold waters: a review of the genus Cuthonella with descriptions of seven new species (Mollusca, Nudibranchia). Contributions to Zoology, 90(2), 216-283.
5) Same with Z. pustulata, some of them are now Z. roginskae
6) Why do you define Cuthona divae as distinct species (accepted in WORMS but not supported in Cella et al. 2016), but synonymize C. hermitophilla with C. nana (according to Cella et al., but in opposition to WORMS)?
7) Please check if all your T. caerulea belong to this species; there are three more species in this complex: T. diljuvia, T. morrowae, T. cuanansis
8) Same with E. rupium, likely a specimen from California should be E. olivaceus (see Cella et al., 2016).
9) Same with Tenellia (Catriona) gymnota, some of specimens should be named as T. aurantia.
Maybe there are some other replacements needed, again, please check Table 1 and your dataset and trees carefully.

Also I did not find any GB numbers of newly sequenced specimens in Table 1.

989-990: how did you understand that cnidosacs of T. nakapila and Tenellia sp. 58 are functional and contain sequestered nematocysts? Did you perform an analysis of their content, at least using the light microscopy? This must be added to M&Ms section.

Validity of the findings

This manuscript is very important since it describes several yet undescribed Indo-Pacific species, and I am fully agree with authors, who stated: "Most taxonomic work has focused on cold-temperate and boreal taxa from the northern hemisphere [...] with little attention directed at the undocumented tropical diversity of the Indian and Pacific Oceans.".
Although I think that the analysed dataset is not complete yet, the validity of their findings and the distinctness of the newly described species is undoubtful. The Discussion section is clear and well organized and provide a fruitful discussion and revision of current taxonomical schemes of the family Fionidae sensu lato. Here I have no major criticism on suggested arguments except those listed above.

I also have some other minor issues:
line 986: Phestilla do have cnidosacs (as well as T. chaetopterana), but these cnidosacs do not contain any sequestered nematocysts (in other words, they are 'empty'), please see Goodheart et al., 2018
line 988-995: it is not clear whether Tenellia nakapila nests within the 'Phestilla' clade or not (again, you must expand your dataset here). If it is sister to this clade, the loss of ability to sequester NCs may be a good synapomorphy of Phestilla, isn't it?
992: what do you mean by 'Clade D'?
995: From your tree I do not see that Phestilla is paraphyletic.

Reviewer 2 ·

Basic reporting

The paper is fundamentally flawed with wrong analysis, biased references and poor unreadable figure with major results. See attached pdf for details.

Experimental design

The taxa selection is highly restricted, disproportional, and clearly not enough for the aims of the present analysis. See attached pdf for details.

Validity of the findings

Analysis is based on previous flawed study. Underlying data are not enough for understanding. Conclusions are poorly stated. Discussion is inconsistent with results. See attached pdf for details.

Additional comments

In the scientific manuscript it is unacceptable to make racist statements on separation of scientists on "developing" and "developed" countries.

Annotated reviews are not available for download in order to protect the identity of reviewers who chose to remain anonymous.

Reviewer 3 ·

Basic reporting

The ms entitled “Stirring up the muck: The systematics of soft-sediment Fionidae (Nudibranchia: Aeolidina) from the tropical Indo Pacific” describes new species belonging to the family Fionidae, increasing our knowledge of the biodiversity of nudibranchs that inhabit the Indo-Pacific Ocean. I think it is a complete study, including the study of the species from an integrative taxonomy with morphological and molecular evidence. The contributions of the authors are of interest to the scientific community, and I consider it appropriate to accept the ms for publication in PeerJ, however, I believe that some changes should be made before being published.
The English of the ms is clear and the references updated. The figures are in general fine, but figure 2. The figure including the phylogenetic tree should be improved, neither species names nor support values can be read.

Experimental design

The experimental design is adequate and complete. The authors explain in detail the methodology followed. However, comments regarding the molecular analysis were included in the PDF. I think it should be improved to increase its clarity.

Validity of the findings

The authors' results are supported using different evidence, including morphological and molecular data. All the data are robust.

Additional comments

In general, the ms is interesting and the research is well done and supported. However, I am a little bit concerned about the final discussion. I think the authors should consider rewriting this section in a more professional and less repetitive way. The authors discuss a problem in this group of animals (and possibly others), where other authors oversplit clades. I agree with them and think it is something that should be discussed. However, I think it is too repetitive and the authors sometimes use words that are not appropriate for a scientific publication, being sometimes redundant and a bit ironic. Like the comment "How many additional species need to be discovered before the authors then decide to divide this into additional genera?". The reader can see the competitiveness, and I think they should better measure the tone of their discussion.
Further comments are included along the text in the PDF file. After reviewing the comments, methodology and modifying the discussion, I believe that the ms can be accepted

Annotated reviews are not available for download in order to protect the identity of reviewers who chose to remain anonymous.

---

## Round 0.2 · Minor Revisions

We are pleased to let you know that the revised version of your manuscript has now passed through the re-review stage and is ready for revision. I am sorry that it has taken longer than normal to receive review for your manuscript. The reviewer provided detailed comments, and I ask that you consider these carefully and correct the errors when revising the manuscript.

Reviewer 1 ·

Basic reporting

no comment

Experimental design

no comment

Validity of the findings

no comment

Additional comments

The authors did very well to review and correct the manuscript and I have no major criticism anymore. However I’ve mentioned a number of minor mistakes/typos/inconsistencies that must be reviewed and corrected before the publication.
Unfortunately, something went completely wrong with Figure 1 displayed in the PDF file for the review. Looks like the upper and the lower parts of the tree lay one upon other and I cannot evaluate modifications made by authors. Please check the submitted files for review.
Anyway, some nodes contain ‘-‘, which likely (?) means that the respective clade was not recovered in ML analysis, but the caption lacks this information, as well as the supplementary Figure S1. Since there are many clades representing an alternative topology in ML, I suggest including the ML tree in the supplementary files as well. Also, on Figure S1 there is ‘Tenellia’ punicea, which must be changed to Tenellia nepunicea.
l. 56-58: likely should be “Korshunova et al., 2017a, 2018, 2023”
l. 71: likely should be ‘contended’
l. 77-78: likely should be Mehrotra et al., 2020, 2024
Here you also say that very little attention is paid to diversity of tropical regions, but the high number of citing literature indicates quite the opposite.
l.94: italise T. yamasui
l. 106: please mention that you studied partial fragments, not whole genes
l. 212-214: you say you studied the radula and jaws, but in the next sentence you indicate that the buccal mass was studied, this is a repetition.
l.250: italise A. turon
l. 253-255: you missed Diaphoraeolis
l. 256: I cannot support an expression ‘the most basal clade’.
l. 268-269: you treat the third group as a clade, but it is not recovered in the ML. A clade cannot be non-monophyletic.
l. 270: why Tenellia ornata is mentioned here separately? I suspect it was also designated as a member of “Trinchesia”
same line: sensu stricto should be italised
l. 271-272: what is the difference between ‘species of the genus ‘Catriona’’ and ‘a clade of the rest of ‘Catriona’’? Please mention species names for clarity. Maybe you could simply say that ‘Catriona’ was not recovered monophyletic due to sister relationships of Tenellia sensu Korshunova et al., 2023 and Catriona gymnota and C. aurantia. Please also include support values for all these clades.
L. 292: missed T. chaetopterana
l. 309-312: I am pretty sure that bPTP analyses oversplitted T. nakapila because of the high intraspecific divergence in the 16S alignment, not because of the unsuccessful COI sequencing
l. 236: regular font for ‘sp. 58’
l774-775: as I’ve already said in my previous review, the representatives of Phestilla (‘corallivorous relatives) do have cnidosacs, but these cnidosacs do not contain nematocysts (Goodheart et al., 2018; Ekimova et al., 2022). So T. chaetopterana is not an exception, because it usually has distal sacs, but these sacs lack nematocysts. Moreover, Putz et al. (2010) have reported on the presence of functional nematocysts in cnidosac of Phestilla lugurbis juveniles. You corrected this issue in the discussion but must be consistent with this in the species description as well.
L783: T. chaetopterana, not T. chaetopterna
I did not find several cited research in the reference list, these are:
1. Folmer et al., 1994
2. Palumbi et al., 1996
3. Colgan et al., 1998
4. Platt et al., 2007
5. Ekimova et al., 2024
Also you have two different papers of Korshunova et al., 2018, please identify which is a or b along the text, Table 1 and in the reference list

---

## Round 0.3 · accepted · Accept

While in production:

L. 750. Please correct T. chaetoptearna to T. chaetopterana.

Insert Palumbi et al., 1996, cited in the text, in the reference list.